# Environmental enrichment enhances patterning and remodeling of synaptic nanoarchitecture as revealed by STED nanoscopy

Waja Wegner[1,2], Heinz Steffens[1,2], Carola Gregor[3,4,5], Fred Wolf[5,6,7], Katrin I Willig[1,2,5]*

[1]Optical Nanoscopy in Neuroscience, Center for Nanoscale Microscopy and Molecular Physiology of the Brain, University Medical Center Göttingen, Göttingen, Germany; [2]Max Planck Institute for Multidisciplinary Sciences, City Campus, Göttingen, Germany; [3]Department of NanoBiophotonics, Max Planck Institute for Multidisciplinary Sciences, Göttingen, Germany; [4]Department of Optical Nanoscopy, Institut für Nanophotonik Göttingen e.V., Göttingen, Germany; [5]Cluster of Excellence "Multiscale Bioimaging: from Molecular Machines to Networks of Excitable Cells" (MBExC), University of Göttingen, Göttingen, Germany; [6]Max Planck Institute for Dynamics and Self-Organization, Göttingen, Germany; [7]Göttingen Campus Institute for Dynamics of Biological Networks, Göttingen, Germany

**Abstract** Synaptic plasticity underlies long-lasting structural and functional changes to brain circuitry and its experience-dependent remodeling can be fundamentally enhanced by environmental enrichment. It is however unknown, whether and how the environmental enrichment alters the morphology and dynamics of individual synapses. Here, we present a virtually crosstalk-free two-color in vivo stimulated emission depletion (STED) microscope to simultaneously superresolve the dynamics of endogenous PSD95 of the post-synaptic density and spine geometry in the mouse cortex. In general, the spine head geometry and PSD95 assemblies were highly dynamic, their changes depended linearly on their original size but correlated only mildly. With environmental enrichment, the size distributions of PSD95 and spine head sizes were sharper than in controls, indicating that synaptic strength is set more uniformly. The topography of the PSD95 nanoorganization was more dynamic after environmental enrichment; changes in size were smaller but more correlated than in mice housed in standard cages. Thus, two-color in vivo time-lapse imaging of synaptic nanoorganization uncovers a unique synaptic nanoplasticity associated with the enhanced learning capabilities under environmental enrichment.

*For correspondence:
kwillig@mpinat.mpg.de

**Competing interest:** The authors declare that no competing interests exist.

## Editor's evaluation

Synapses mediate information transmission in the brain, and part of the synaptic structure called spines are the receiving end of signal transfer between neurons. Using a custom-built superresolution microscope, the study reveals the nanoscale structural dynamics of individual spine shape and its resident scaffolding protein PSD95 simultaneously, in mouse cortex in vivo. Aspects of the structural dynamics are found to differ depending on whether mice have been reared in a simple housing or in an enriched environment, the latter condition being associated with enhanced activity.

## Introduction

### Activity-driven changes of spine dynamics

Over an entire lifespan, cognitive, sensory, and motor learning is associated with changes to a specific assembly of synapses which is often termed the memory engram (*Poo et al., 2016*). Thus, spines emerge, disappear, or change with cellular processes underlying learning, and even 'remember' previous sensory experience (*Hofer et al., 2009*). There is also evidence that learning induces structural and functional synaptic changes similar to long-term potentiation (LTP) protocols (*Poo et al., 2016*). In this concept of learning, the maintenance of memory critically depends on the stability of the underlying synaptic connections. Synaptic structures, however, are highly volatile intrinsically as such that synaptic connections undergo continuous spontaneous remodeling without any activity (*Mongillo et al., 2017*; *Ziv and Brenner, 2018*); for example, intact synapses are formed despite abolishing pre-synaptic release (*Sigler et al., 2017*) or network silencing (*Hazan and Ziv, 2020*). Due to spatial resolution constraints, previous in vivo studies mostly focused on the persistency of spines and synapses in terms of their appearance and elimination; the spine size was estimated from fluorescence intensity. Therefore, directly assessed activity-driven changes in synapse or spine head size in vivo are missing.

### Spine to synapse structural correlation

Decades of neuroscience research has shown a tight correlation between structural plasticity, the morphological transformation of spines and synapses, and modifications in synaptic transmission, termed functional plasticity (*Yuste and Bonhoeffer, 2001*), which was confirmed recently at ultrastructural resolution (*Holler et al., 2021*). Functional changes are linked to anchoring amino-3-hydroxy-5-methyl-4-isoxazolepropionic acid receptors (AMPAR) via transmembrane AMPAR-regulatory proteins (TARPs) to the post-synaptic density (PSD) scaffolding protein PSD95 (*Compans et al., 2016*; *Herring and Nicoll, 2016*). AMPAR-mediated currents were shown to increase simultaneously with the spine head directly after LTP induction. Recent evidence suggests that not all synaptic structures follow this fast dynamic; as such, accumulations of PSD95 increase in vitro with a delay of ~1 hr after inducing LTP, whereas the spine heads increase in less than a minute (*Bosch et al., 2014*; *Meyer et al., 2014*). Thus, the relation between spine size and PSD organization (*Arellano et al., 2007*; *Cane et al., 2014*; *Harris et al., 1992*; *Meyer et al., 2014*) may be temporally dynamic and complex during synaptic change and the remodeling of PSD and spine head plasticity might be more independent than previously thought. Here, we investigate the structural correlation at nanoscale of the spine and post-synapse in vivo using mice reared in enriched environment representing increased activity conditions and normal housing representing baseline conditions.

### PSD95 nanoorganization

By applying superresolution microscopy we and others have shown that PSD95 is often arranged in clusters, rings, or a more complex nanopattern, and that this nanoorganization undergoes pronounced intrinsic structural changes on the time scale of minutes to hours (*Hruska et al., 2018*; *MacGillavry et al., 2013*; *Wegner et al., 2018*). Recently, it was shown that such a sub-synaptic nanoorganization exists at the pre- and post-synaptic site and that both are aligned in the so-called nanocolumns (*Hruska et al., 2018*; *Tang et al., 2016*). Computational studies show that changes of the shape of the PSD (*Franks et al., 2003*) or modest clustering of AMPAR (*Savtchenko and Rusakov, 2014*) is a highly efficient way to modify synaptic strength without altering the total amount of receptors. Moreover, AMPAR current amplitude drops significantly already at 50 nm offset between pre-synaptic glutamate release site and AMPAR cluster (*Haas et al., 2018*). Therefore, activity-driven structural rearrangement at the nanoscale to align post-synaptic receptors to pre-synaptic release sites may be faster than the incorporation of new molecules to induce changes in the synaptic strength.

### This study

Here, we apply nanoscale stimulated emission depletion (STED) microscopy to map temporal changes of the spine head and PSD95 accumulations simultaneously in vivo, and address (1) the plasticity of spine heads and synapses at baseline at time scales similar to LTP processes; (2) the correlation between PSD95 and spine head size changes; (3) the plasticity of the PSD95 nanoorganization; and (4) whether enhanced activity modifies the structure and/or plasticity of these measures. For this

purpose, we employ environmental enrichment (EE) where a mouse is provided with multi-sensory stimulation, cognitive activity, social interactions, and physical exercise, which modifies the degree of plasticity and dynamics of cortical sensory circuits. EE has been shown to accelerate the maturation of new neurons, increase pre- and post-synaptic protein levels (*Nithianantharajah et al., 2004*), and facilitate synaptic plasticity (*Greifzu et al., 2014*; *Nithianantharajah and Hannan, 2006*). Structural changes include increased dendritic branching and spine density (*Gelfo et al., 2009*; *Leggio et al., 2005*) and induction of spine formation (*Yang et al., 2009*). However, it is unknown whether the effects of EE leave their mark only at the level of spine formation and elimination such as observed by *Yang et al., 2009*, or whether EE also affects the dynamics and nanostructure of all individual spines and PSDs.

To unambiguously assess these relationships between spine morphology and PSD95, we set up a virtually crosstalk-free two-color STED microscope for in vivo superresolution imaging of EGFP and Citrine. We utilized a transcriptionally regulated antibody-like protein to visualize endogenous PSD95 without introducing overexpression artifacts (*Gross et al., 2013*). Together with a membrane label, we simultaneously recorded temporal morphological changes of the spine head and PSD95 nanoorganization and compared its dynamics between mice raised in EE and in standard cages (control, Ctr). By investigating the synaptic dynamics in the mouse visual cortex, a brain region known to exhibit enhanced plasticity under EE conditions (*Baroncelli et al., 2010*; *Kalogeraki et al., 2017*) and accessible to light microscopy, we demonstrate that spine head and PSD95 size distributions decrease in variability while spine heads increase in average size under EE. In addition, PSD95 underwent much stronger directional changes in size and reorganization of their nanoorganization under EE, while the average change in amplitude was smaller compared to the controls. Dynamical changes of PSD95 and spine head size correlated only mildly.

## Results

### Virtually crosstalk-free two-color STED Microscope

To unambiguously dissect the dynamics of distinct synaptic components and examine their interactions, we established a virtually crosstalk-free two-color STED microscope for imaging of EGFP and EYFP or Citrine, respectively. The challenge for in vivo two-color STED microscopy is to find an in vivo compatible pair of fluorescent molecules with similar emission spectra so that it can be depleted with the same STED beam, but at the same time can be temporally or spectrally separated (*Willig et al., 2021*). Previous attempts with STED microscopy of EGFP and EYFP utilized a single excitation wavelength and two-color detection, which suffered from high crosstalk, and therefore required a linear unmixing of the channels (*Tønnesen et al., 2011*). Channel unmixing, however, requires large signal-to-noise levels. To reduce crosstalk and thus avoid the necessity of channel unmixing, we extended our previously described in vivo STED microscope (*Willig et al., 2014*) by an additional two-color excitation and detection to selectively excite the green or yellow fluorescent protein (*Figure 1A*). We utilized 483 nm pulses to excite EGFP and detected its spontaneous fluorescence at 498–510 nm (Det1, *Figure 1B*). EYFP (or Citrine) was excited at 520 nm and detected mainly at 532–555 nm (Det2, *Figure 1B*). The 483 and 520 nm excitations were switched on and off alternately while recording two consecutive line scans to temporally separate the EGFP and EYFP/Citrine emissions. To determine the crosstalk between both channels, we investigated one-color labeled cells expressing EGFP or EYFP. Live-cell imaging showed that we have designed a virtually crosstalk-free microscope with only ~8% crosstalk of EYFP in the EGFP channel (channel 1) and ~5% of the EGFP signal in the EYFP channel (channel 2) which does not require linear unmixing (*Figure 1C*). STED was performed by a 595 nm laser beam passing a vortex phase plate to create a donut-shaped intensity profile in the focal plane of the objective as previously described (*Eggeling et al., 2015*).

### Two-color in vivo STED microscopy of endogenous PSD95 and spine morphology

Having established the virtually crosstalk-free detection of two STED approved fluorescent proteins (*Nägerl et al., 2008*; *Willig et al., 2006*), we set out to simultaneously superresolve the nanoorganization of the PSD protein PSD95 and the associated spine morphology in vivo. To this end, we generated recombinant adeno-associated viral (AAV) particles encoding fusion proteins under control of the human

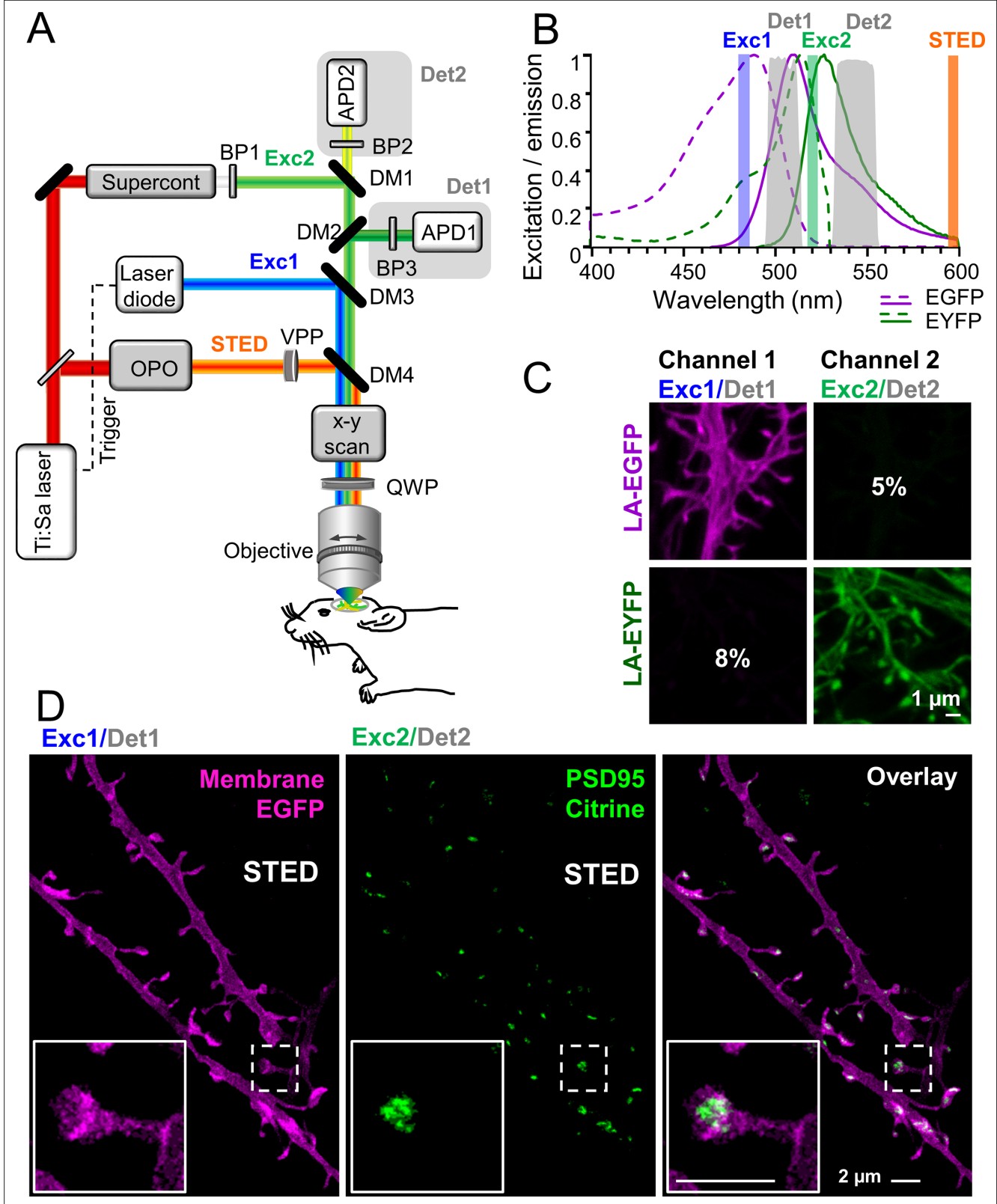

**Figure 1.** Virtually crosstalk-free two-color in vivo stimulated emission depletion (STED) microscopy. (**A**) Custom-designed in vivo STED microscope with pulsed 483 nm (Exc1) and 520 nm (Exc2) excitation and 595 nm STED laser. APD: avalanche photon detector, BP: bandpass filter, Det: detection, DM: dichroic mirror, OPO: optical parametric oscillator, QWP: quarter wave plate, VPP: vortex phase plate. See Material and methods section for details. (**B**) Excitation (dashed line) and emission (solid line) spectrum of EGFP and EYFP and wavelength regions for selective excitation and detection. (**C**)

*Figure 1 continued on next page*

*Figure 1 continued*

Cultured hippocampal neurons expressing the actin marker Lifeact (LA)-EGFP or LA-EYFP. Excitation of EGFP at 483 nm close to its excitation maximum (**B**) and detection at 498–510 nm (Det1) reduced the crosstalk to 5% in channel 2 (**C**). EYFP or Citrine was excited at 520 nm close to its maximum (**B**) and detected at 532–555 nm (Det2) which resulted in a low crosstalk of 8% in channel 1 (**C**). (**D**) In vivo STED microscopy image of apical dendrite in layer 1 of the visual cortex of an anesthetized mouse. Labeling of membrane (myr-EGFP-LDLR(Ct)) and PSD95 (PSD95.FingR-Citrine) visualized the spine morphology and PSD95 nanoorganization at superresolution (**D**, inset). Images are smoothed and represent maximum intensity projections (MIPs). No unmixing was performed due to the low crosstalk.

The online version of this article includes the following figure supplement(s) for figure 1:

**Figure supplement 1.** Resolving capability of the custom-built two-color in vivo stimulated emission depletion (STED) microscope.

Synapsin promoter (hSyn). To visualize PSD95, we expressed the antibody-like protein PSD95.FingR, which has been shown to label endogenous PSD95 (*Gross et al., 2013*; *Willig et al., 2021*) attached to the fluorescent protein Citrine. The expression of PSD95.FingR-Citrine is controlled by a transcriptional regulation system that prevents expression after saturation of the binding sites and therefore reduces background (*Gross et al., 2013*). For spine morphology, we expressed myr-EGFP-LDLR(Ct), a combination of a myristoylation site (myr), EGFP, and the C-terminal (Ct) cytoplasmic domain of low-density lipoprotein receptor (LDLR), a potent marker for the dendritic membrane (*Kameda et al., 2008*; *Willig et al., 2021*). In order to reduce the density of labeled neurons and thus overlapping fluorescence signals, the fluorescent labels were incorporated into AAVs in reverse orientation in the vector between double-floxed inverted open reading frames (DIO). Expression of the labels was enabled by Cre recombinase encoding AAVs co-injected at low concentration (*Willig et al., 2021*). Hence, the density of labeled neurons was tuned by the dilution of the Cre expressing AAV and the brightness of the PSD95 and membrane labeling by the concentration of the respective AAVs; thus the density of labeled cells and their brightness were independent variables. We co-injected the three AAVs into pyramidal cell layer 5 of the visual cortex. Three to six weeks after transduction, the mice were anesthetized and a cranial window was inserted over the visual cortex. To perform motion and aberration-free imaging at nanoscale resolution, the cranial window needs to be of highest quality. As described in detail in *Steffens et al., 2020*, critical steps involve a craniotomy that is as atraumatic as possible and does not damage the cortical surface when drilling or removing the bone plate and dura mater. Moreover, the gap between the brain surface and the cover glass needs to be negligibly small and the right choice of the dental cement is important to avoid bending of the cover glass. With such an optimized cranial window, STED microscopy visualized dendrites, spines, and attached PSD95 assemblies in layer 1 of the visual cortex at superresolution (*Figure 1C*) with a resolution of at least ~70 nm for Citrine and ~84 nm for EGFP (*Figure 1—figure supplement 1A*, B). The inset in *Figure 1D* reveals a perforated nanoorganization of endogenous PSD95 that would not be detectable with conventional in vivo light microscopy due to the insufficient resolution. Due to the low crosstalk, a computational post-processing such as unmixing was not required. We observed mainly dendrites that expressed both, EGFP and Citrine, and only rarely found dendrites expressing either Citrine or EGFP alone.

## EE housed mice exhibit less variability in the size of spine head and PSD95 assemblies while their heads are larger than in control

Having established the two-color in vivo STED microscopy, we addressed the question of whether spines or PSDs exhibit systematic differences in size between EE and Ctr mice. EE mice were housed in a large, two-floor cage equipped with three running wheels for physical exercise, a labyrinth for cognitive stimulation, a tube and a ladder to change between the two levels, and were kept in groups of up to 12 female mice to allow manyfold social interactions (*Figure 2—figure supplement 1A–C*). Ctr mice were raised in pairs in standard cages without any equipment (*Figure 2—figure supplement 1D, E*). The mice were transduced with AAVs as described above and 3–6 weeks after transduction, they were anesthetized and a cranial window was implanted above the visual cortex. Imaging commenced about 2.5 hr after onset of the anesthesia. We acquired two-color z-stacks mainly of dendrites, which were in parallel to the focal plane. To analyze differences in morphology of the dendritic spines, we encircled the spine head and PSD95 at their largest extent and computed the respective area (*Figure 2A*, see Materials and methods section for details). We only considered spines containing a PSD95 label and therefore most likely form functional synapses. Occasionally we

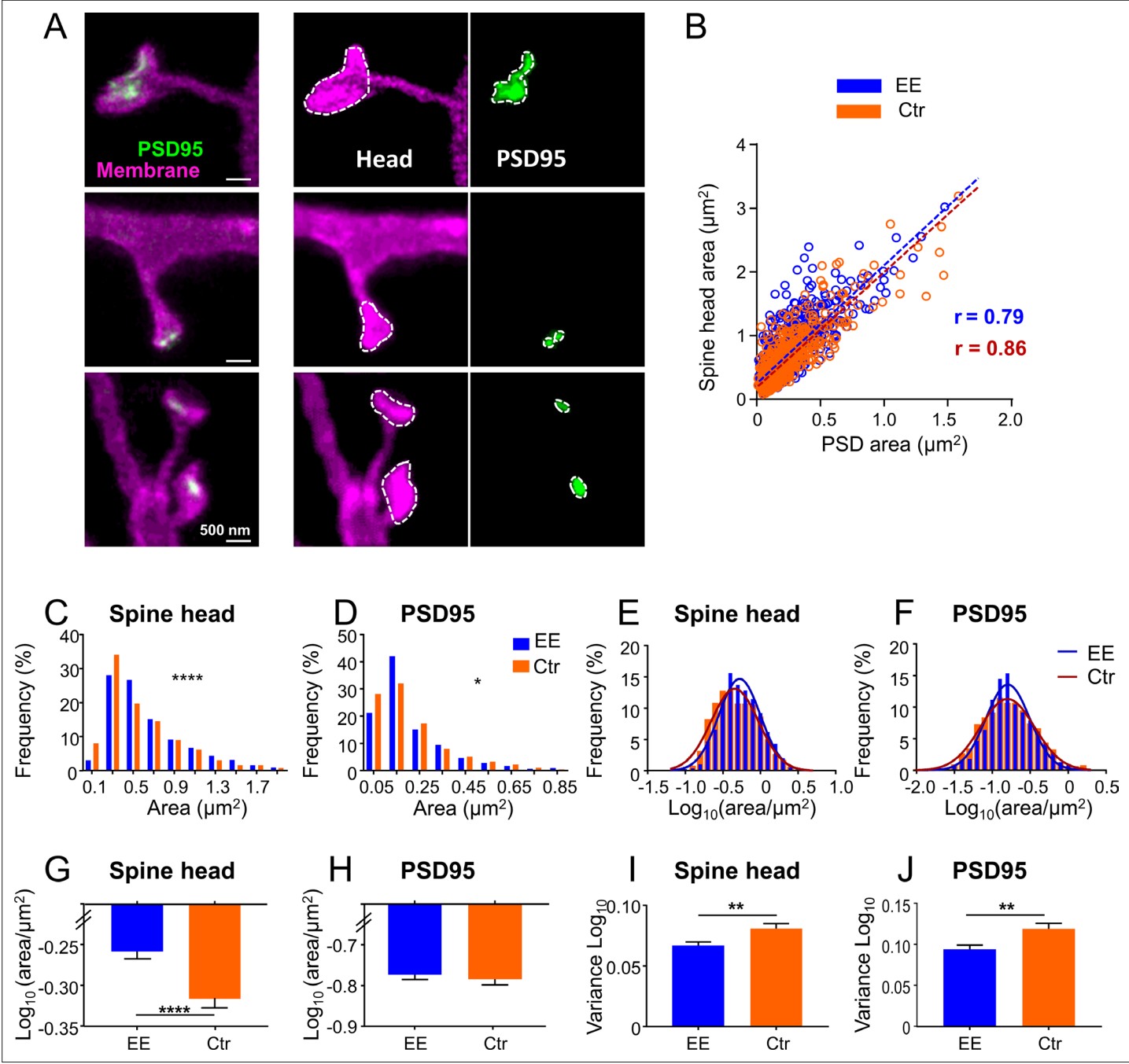

**Figure 2.** Size distributions of spine head and PSD95 area are sharper and show larger heads for mice housed in environmentally enriched (EE) than in standard (control [Ctr]) cages. (**A**) Stimulated emission depletion (STED) images of dendritic spines (magenta) and associated PSD95 assemblies (green). Images are smoothed; maximum intensity projection (MIP) (left), contrast enhanced images for area analysis (middle, right). Spine heads (middle) and PSD95 assemblies (right) were encircled to compute the area. (**B**) Strong correlation of absolute spine head and PSD95 area in Ctr (orange) and EE (blue) housed mice. Linear regression lines are dashed and Pearson's correlation coefficient r is displayed (EE and Ctr, deviation from zero: p < 0.0001). (**C, D**) Frequency distributions of spine head area (**C**) and PSD95 area (**D**) are positively skewed and significantly different between EE and Ctr housed mice (Kolmogorov-Smirnov test, C: ****p < 0.0001, D: *p = 0.013). Graphs display center of BIN, single large values are cut off. (**E, F**) Same data as shown in (**C, D**), but logarithmic values. Solid lines represent Gaussian functions fitted to the respective histogram. (**G–J**) Mean (G: $EE_{spine}$: –0.259, $Ctr_{spine}$: –0.317, H: $EE_{PSD}$: –0.771, $Ctr_{PSD}$: –0.781) and variance (**I, J**) of logarithmic data shown in (**E, F**) + SEM (unpaired t-test with Welch's correction: G: ****p < 0.0001, H: p = 0.54, I: **p = 0.006, J: **p = 0.003). Number of analyzed mice and spines: EE: 4x ♀-mice, $n_{Spine/PSD95}$ = 795; Ctr: 4x ♀-mice, $n_{Spine/PSD95}$ = 634.

The online version of this article includes the following source data and figure supplement(s) for figure 2:

**Source data 1.** Spine head and PSD95 assembly sizes.

*Figure 2 continued on next page*

*Figure 2 continued*

**Figure supplement 1.** Housing conditions for enriched environment (EE) and control (Ctr) mice.

**Figure supplement 2.** Correlation between PSD95 area and brightness; spine density.

observed long, thin spines without a head, sometimes called filopodia in the literature, which were not analyzed. The spine head area correlated with the PSD95 area for EE and Ctr (*Figure 2B*), corroborating EM and two-photon microscopy studies (*Arellano et al., 2007*; *Cane et al., 2014*; *Meyer et al., 2014*). The histograms of spine head (*Figure 2C*) and PSD95 area (*Figure 2D*) were positively skewed and the distributions were significantly different between EE and Ctr mice. We determined a median spine head area for EE housed mice of 0.527 (interquartile range [IR]: 0.359–0.844) $\mu m^2$ and 0.462 (IR: 0.290–0.787) $\mu m^2$ for Ctr mice. The area of PSD95 on the same set of spine heads was 0.158 (IR: 0.107–0.269) $\mu m^2$ in median for EE mice and 0.161 (IR: 0.092–0.286) $\mu m^2$ for Ctr mice which is in accordance with previous EM studies reporting 0.15 $\mu m^2$ for the PSD area in the mouse visual cortex (*Harris and Weinberg, 2012*). This size of the PSD95 area of layer 5 pyramidal neurons is slightly larger than our previously reported diameter of 354 nm which corresponds to ~0.10 $\mu m^2$ for a circular distribution, obtained in a ubiquitously expressing PSD95-EGFP knock-in mouse (*Wegner et al., 2018*); therefore, the larger size of the PSD95 area could reflect the larger size of the spine heads of layer 5 pyramidal neurons (*Konur et al., 2003*). To further dissect these differences, we next plotted the histograms of the logarithm of spine head (*Figure 2E*) and PSD95 areas (*Figure 2F*). All four histograms (*Figure 2E and F*) are symmetric and well described by a Gaussian function. This is in line with prior work showing that the spine and PSD95 fluorescence is log-normally distributed in general, which is predicted by multiplicative processes of ongoing spine plasticity (*Hazan and Ziv, 2020*; *Loewenstein et al., 2011*). The Gaussian function fitting the spine head area is shifted to the right and is narrower for EE housed mice (*Figure 2E*). This is manifested by a significantly larger spine head area (*Figure 2G*) and smaller variance of the size distribution (*Figure 2I*) of EE housed mice, which might imply a preferential loss or adaptation of small spines (*Figure 2E*). The distributions of the logarithm of the PSD95 area (*Figure 2F*) are almost centered and thus the average area is not significantly different between EE and Ctr mice (*Figure 2H*). However, the variance (*Figure 2J*) is smaller for EE housed mice which is also visible in the Gaussian distribution that is narrower for EE housed mice (*Figure 2F*), that is, having less extreme values. Since previous studies often used the brightness of PSD95 assemblies or spines, respectively, as a measure of size, we also analyzed the PSD95 brightness and found a coefficient of determination $R^2 = 0.76$ between the brightness and nanoscale size (*Figure 2—figure supplement 2A*), indicating that the brightness is a deficient correlate of synaptic size. We also analyzed the spine density which was not significantly different between EE and Ctr housed mice (*Figure 2—figure supplement 2B*). These results show that EE leaves a trace in spine head size and variability of the post-synaptic size, both of which have been shown to correlate with synaptic strength (*Holler et al., 2021*).

## Weak correlation of PSD95 and spine head area changes over minutes to hours

We showed above a strong correlation between the size of PSD95 assemblies and the spine head (*Figure 2B*) which corroborates previous findings of close structural correlation between PSD and spine size (*Arellano et al., 2007*). However, on which time scale the dynamic changes between these two features are linked in vivo has remained unknown. We thus asked whether and how temporal changes of these parameters were correlated. Therefore, we performed time-lapse STED microscopy of spine morphology and PSD95 for EE and Ctr housed mice for up to 4 hr as described above and analyzed temporal changes of these parameters (*Figure 3A–L*). Imaging commenced ~2.5 hr after onset of the anesthesia and the time-lapse was recorded at baseline, without a stimulation. We recorded STED images at different fields of view (FOV); each FOV was recorded for three time points at a time interval Δt of either 30 min (*Figure 3A and B*), 60 or 120 min. Over these time periods the spines were mostly stable. Occasionally, a spine was lost or a new one appeared; none of these dynamic spines carried PSD95 and they were therefore most likely highly dynamic filopodia (*Berry and Nedivi, 2017*). For each time point, the size of the spine head and PSD95 assembly was analyzed as described above. The average spine head area and PSD95 area did not show large changes over the time period of 120 min (*Figure 3C*). Thus, we did not observe directed changes as a result of light-induced

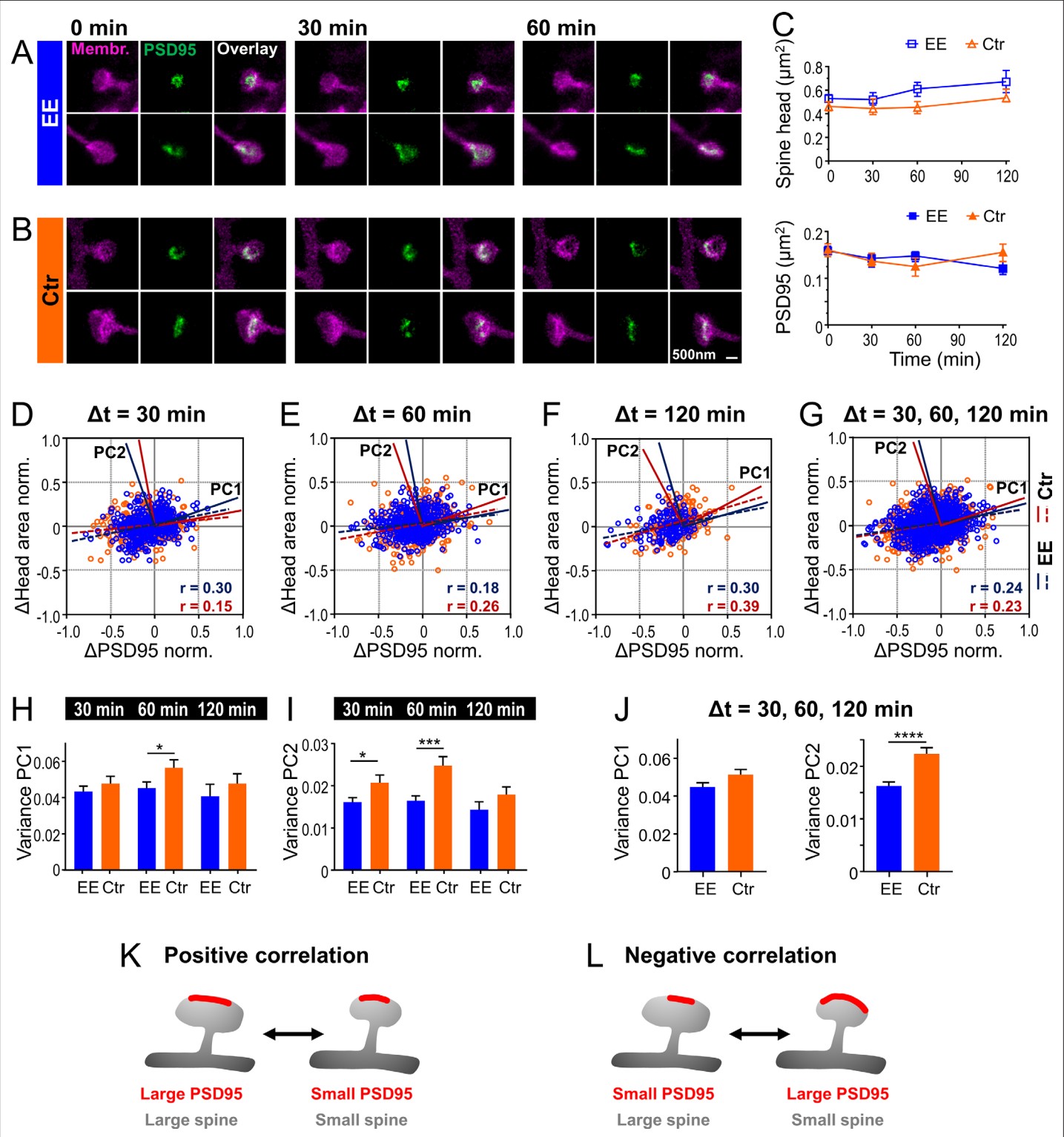

**Figure 3.** Temporal changes of spine head and PSD95 area are weakly positively correlated with a larger variability for control (Ctr) than environmental enrichment (EE) housed mice. (**A, B**) Representative sections of spine heads and corresponding PSD95 of time-lapse in vivo two-color stimulated emission depletion (STED) microscopy images for EE (**A**) and Ctr (**B**) housed mice at time points of 0, 30, and 60 min. Images are smoothed and shown as maximum intensity projection (MIP). (**C**) Median ±95% confidence interval (CI) of spine head and PSD95 areas of Ctr (triangle, orange) and EE (square, blue) housed mice over time. (**D–G**) Normalized changes in spine head and PSD95 area after 30 min (**D**), 60 min (**E**), and 120 min (**F**) time intervals and compiled changes of all time intervals (**G**). Linear regression lines are dashed (deviation from zero: p < 0.0001) and Pearson's correlation coefficient r

*Figure 3 continued on next page*

*Figure 3 continued*

is displayed. Solid lines represent the principal components 1 and 2 (PC1, PC2) of the principal component analysis (PCA) for Ctr (red) and EE (blue), respectively. (**H–J**) Variance along PC1 and PC2 of normalized changes plotted in (**D–G**); variance of PC1 (**H**), PC2 (**I**), and compiled variance over all time intervals (**J**) + SEM (unpaired t-test EE vs. Ctr; H: 30 min: p = 0.36, 60 min: *p = 0.045, 120 min: p = 0.40; I: 30 min: *p = 0.027, 60 min: ***p < 0.001, 120 min: p = 0.17; J: PC1: p = 0.063, PC2: ****p < 0.0001). (**K, L**) Illustration of temporal changes between spine head size and PSD95 area; positive correlation (**K**): growth and shrinkage of spine head size and area of PSD95 assemblies goes hand-in-hand; negative correlation (**L**): a growing PSD95 assembly on a shrinking spine and vice versa (**K**). (**C–J**) Number of analyzed mice and spines: 4x ♀-mice for EE and 4x ♀-mice for Ctr; number of spines with PSD95 assemblies: EE, t = 0: $n_{Spine/PSD95}$ = 795, t = 30 min: 326, t = 60 min: 388, t = 120 min: 151; Ctr, t = 0: 634, t = 30 min: 233, t = 60 min: 285, t = 120 min: 189. Time intervals are pooled; for example, Δt = 30 min includes 0–30 and 30–60 min.

The online version of this article includes the following source data and figure supplement(s) for figure 3:

**Source data 1.** Spine head and PSD95 assembly sizes for each time point.

**Figure supplement 1.** Temporal changes in spine head and PSD95 area over time course.

stimulation or phototoxicity and neither observed blebbing. We computed normalized size changes over time so that both, positive (growth) and negative (shrinkage), changes were symmetric with a boundary value of ±1. *Figure 3D–G* shows scatter plots of the normalized changes of the spine head over changes of the PSD95 area for each time interval (*Figure 3D–F*) and cumulated changes of all time intervals (*Figure 3G*); the data points are scattered over all four quadrants (for percentage changes refer to *Figure 3—figure supplement 1A–C*). A linear regression analysis revealed a weak but significant positive correlation (r between 0.15 and 0.39, *Figure 3D–G*), such that an increase of PSD95 area is more frequently accompanied by an increase in spine head area and vice versa (*Figure 3K*). Some data points, however, also represented anti- or negatively correlated changes, that is, shrinking PSD95 areas on a growing spine or vice versa (*Figure 3L*). Interestingly, PSD95 assemblies displayed larger normalized changes in size than spine heads, which is evident from the elliptic distributions in *Figure 3D–G* or the median percent changes (*Figure 3—figure supplement 1D*, E); spine heads grew by ~20% and shrunk by ~15% while PSD95 assemblies grew by ~20–25% and shrunk by 25–30%. To further quantify the portion of correlated and anti-correlated changes, we performed a principal component analysis (PCA). Principal component 1 (PC1) is by definition the axis along which the data shows the maximum of variance and the eigenvalue of PC1 is the variance along this axis. The PC1 vector was for all three time intervals in the first (and third) quadrant (*Figure 3D–F*) and therefore attributing for positively correlated changes. The second principal component (PC2) is by definition perpendicular to PC1 and therefore in the second (and fourth) quadrant (*Figure 3D–F*), attributing for negatively correlated changes. The variance of PC1 and thus correlated changes is ~3 times larger at all time intervals than that of PC2, that is, anti-correlated changes (*Figure 3H, I*; *Figure 3—figure supplement 1F*). Thus, we found besides the positively correlated changes also a large portion of anti-correlated changes reflecting a temporal uncoupling of spine head and PSD95 morphology. For all time intervals, the variance of PC1 and PC2 was higher for Ctr housed mice, although not always significant (*Figure 3H, I*). To compare Ctr versus EE housed mice, we averaged the variance over all time intervals (*Figure 3J*) as the changes did not vary significantly within one group. Thus, the variance of Ctr housed mice was statistically significantly higher for PC2 but not for PC1. This indicates that negatively correlated changes contribute much less in the EE housed mice indicating a stronger, positive coupling between changes in spine head size and PSD95 area.

In summary, the synaptic structure is highly dynamic and spine heads and PSD95 assemblies change in size by >20% already after 30 min. The variance of these changes is smaller and positively correlated changes are more pronounced for EE housed mice. These results suggest that EE housed mice undergo smaller but more directed morphological changes under anesthesia.

## Multiplicative downscaling of PSD95 area is different between EE and Ctr housed mice

Since our parameters were log-normally distributed, indicating a multiplicative dynamic, we asked whether the temporal changes were also regulated by multiplicative processes (*Hazan and Ziv, 2020*; *Loewenstein et al., 2011*). Thus, we plotted the synaptic size after different time intervals Δt as a function of the original size (*Figure 4A–F*). A linear regression analysis revealed a slope <1 and positive y-intercept for spine head and PSD95 area, for both EE and Ctr housed mice. Plotting the size difference between the different time points as a function of the original size indicates that large spines

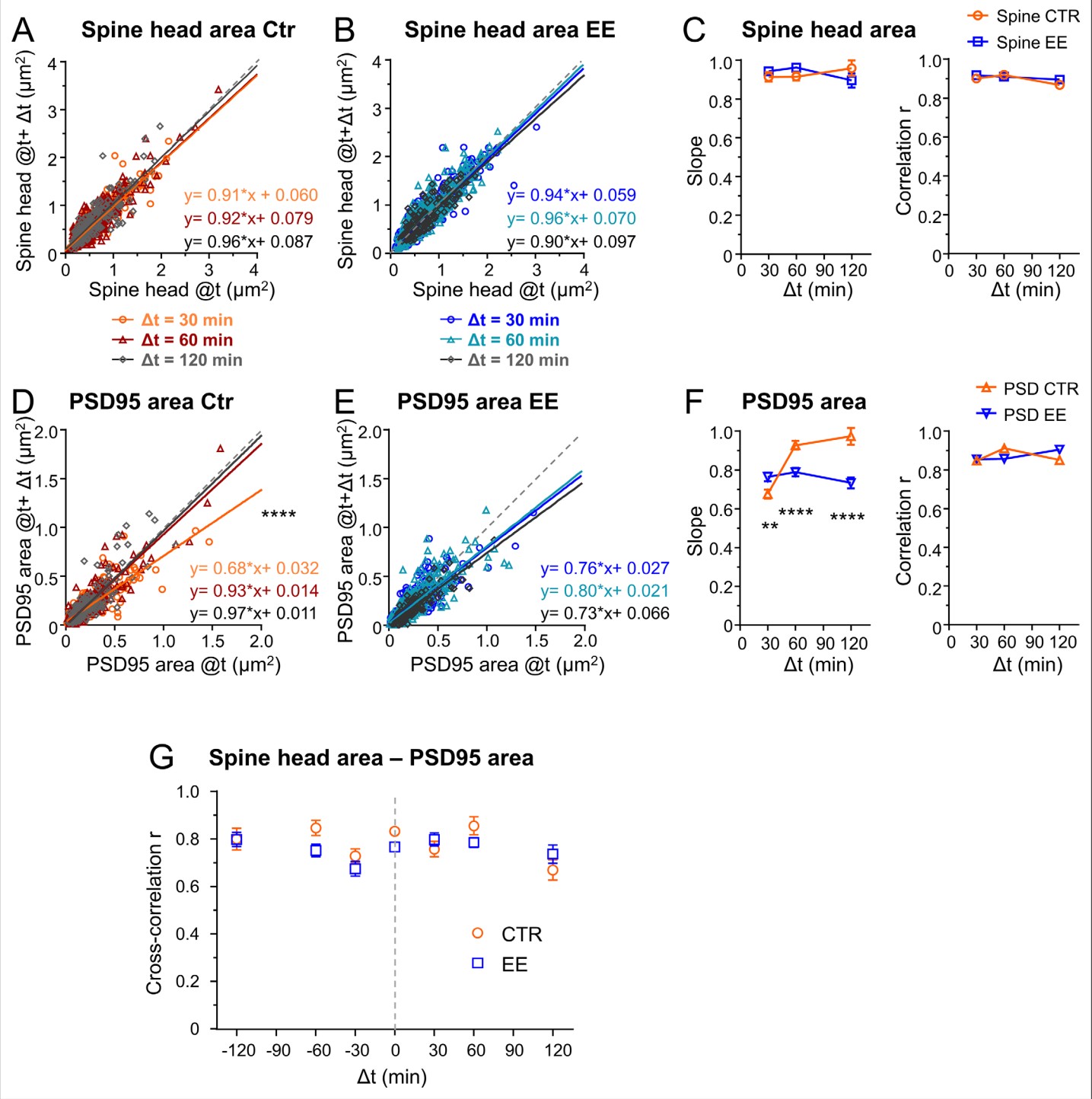

**Figure 4.** Environmental enrichment (EE) housed mice show an increase in multiplicative downscaling of PSD95 area over control (Ctr). (**A–F**) Spine head and PSD95 area after different time intervals Δt of 30, 60, and 120 min as function of their initial area at time t (**A, B, D, E**). Solid lines show linear regression fits of the displayed equation; the identity line is dashed (analysis of covariance; are slopes equal? A: p = 0.53, B: p = 0.29, D: ****p < 0.0001, E: p = 0.39). (**C**) Slope ± SE of fit to spine head changes (**A, B**) (left, are slopes different? EE vs. Ctr: 30 min: p = 0.37, 60 min: p = 0.11, 120 min: p = 0.26) and Pearson's correlation r (right) of linear regression. (**F**) Slope ± SE of fit to PSD95 area changes (**D, E**) (left, are slopes different? EE vs. Ctr: 30 min: **p < 0.01, 60 min: ****p < 0.0001, 120 min: ****p < 0.0001) and r value (right). (**G**) Cross-correlation between spine head and PSD95 area for EE and Ctr housed mice. Error bars are bootstrap ± SD. Same data set as in *Figure 3*; the same time intervals Δt are pooled.

The online version of this article includes the following figure supplement(s) for figure 4:

**Figure supplement 1.** Size changes of environmental enrichment (EE) and control (Ctr) housed mice.

tend to shrink and small spines tend to grow; this becomes evident as the changes are rather negative for larger sizes and positive for smaller sizes (*Figure 4—figure supplement 1*). Such a tendency is often called regression to the mean and is a frequently observed statistical phenomenon. However, it is driven by biological processes, and the strength of those changes may vary under different conditions such as between EE and Ctr. To quantify these changes in synapse and spine head size, we use a Kesten process which was recently applied as a model for spine dynamics and their skewed size distribution. In this model a noisy multiplicative downscaling is combined with a noisy additive term (*Hazan and Ziv, 2020*; *Statman et al., 2014*). Thus, the negative slope as shown in *Figure 4—figure supplement 1* which is equal to 1-slope of *Figure 4* would be regarded as a time-dependent multiplicative downscaling factor. We observed a significantly different slope, that is, multiplicative downscaling for the PSD95 area between EE and Ctr housed mice (*Figure 4F*). While the slope was constant or decreased slightly over time for EE housed mice, it rose up to nearly one for Ctr mice. Interestingly, this effect was specific for PSD95 and was not observed for the spine head size (*Figure 4A–C*); the slope for spine head size changes was rather constant over time and similar between EE and Ctr mice. In summary, we found a significant increase in downscaling over time for PSD95 in EE housed mice as compared to Ctr after 60 and 120 min.

## Despite fluctuations, spine heads and PSD95 assemblies are stable on average over 2 hr

Next, we asked how these large temporal changes influence the stability of the synapse. Do growing spines continue to grow and do shrinking spines continue to shrink? Thus, we computed the Pearson's correlation coefficient r between the different time points (*Figure 4C and F*, right). The r value was relatively constant over all time intervals and not significantly different between spine heads and PSD95 assemblies or EE and Ctr house mice. As such we did not find changes in *r* or its corresponding coefficient of determination $R^2$ value such as described in vitro (*Hazan and Ziv, 2020*) at our time window of up to 2 hr. This indicates that large spine heads tend to stay rather large and smaller ones stay small.

## No temporal shift between spine head and PSD95 area changes at baseline

We found a strong correlation coefficient of ~0.8 between spine head and PSD95 area (*Figure 2B*). However, if the PSD expands with a temporal delay of ~1 hr to the spine head as suggested by the work of *Bosch et al., 2014*; *Meyer et al., 2014*, the correlation should be even higher when comparing spine heads and PSD95 area at different time points. Therefore, we computed the cross-correlation between these measures for all time intervals. *Figure 4G* shows that the cross-correlation between the PSD95 and spine head area was ~0.8 over all time intervals of 0–120 min and thus we did not observe directional changes of these parameters at baseline. In particular, we did not observe that PSD95 increase would systematically succeed a spine head increase or vice versa as both would result in an asymmetric cross-correlation.

## PSD95 nanoorganization changes faster in EE mice

Previous EM studies have shown that PSDs on large spines, often called mushroom spines, are frequently perforated (*Stewart et al., 2005*). In our previous publication (*Wegner et al., 2018*), we investigated PSD95 morphology in homozygous knock-in mice and showed that PSD95 assemblies on large spines were often perforated and appeared ring-like or clustered. This nanoorganization was highly dynamic and changed in shape within a few hours. Now, we asked whether these changes would be different for mice housed in EE. *Figure 5A* shows examples of two-color STED images of perforated PSD95 assemblies and their associated spine revealing temporal changes in the PSD95 nanopattern. The nanopattern was similar to that of the knock-in mice (*Wegner et al., 2018*) and is quite complex; we often observed clusters but also continuous structures of horseshoe or more complex shapes. Previous studies have counted number of nanodomains for these structures (*Hruska et al., 2018*). However, in our opinion, this does not satisfactorily reflect the complexity of the structure and thus we performed a visual inspection analogous to *Wegner et al., 2018*. Morphological changes were categorized for each time point as no change, subtle change, or strong change with respect to the first observation at t = 0 by three independent persons (*Figure 5B and C*). Ctr housed mice showed a

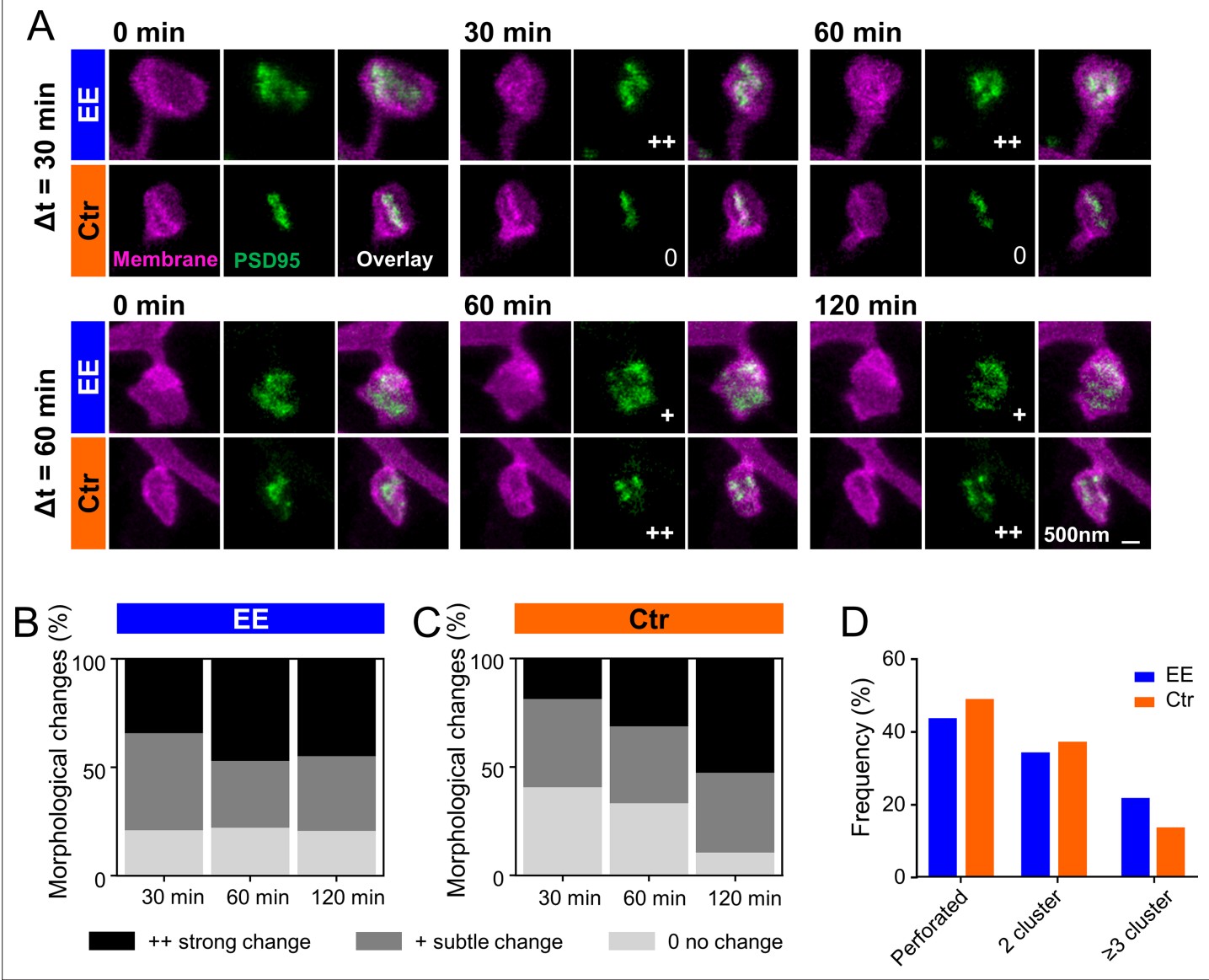

**Figure 5.** PSD95 nanopattern is different between environmental enrichment (EE) and control (Ctr) and changes faster for EE housed mice. (**A**) Sections of two-color stimulated emission depletion (STED) images (smoothed, maximum intensity projection [MIP]) of EE and Ctr housed mice at the indicated time points showing the spine membrane (magenta) and PSD95 (green). Morphological change is indicated with: 0 = no change; + = subtle change; ++ = strong change. (**B, C**) Stacked histogram of the relative frequency of morphological changes in PSD95 nanopattern in EE (**B**) and Ctr (**C**) housed mice; all changes refer to t = 0 min. (**D**) Morphometry of PSD95 nanopattern of EE and Ctr housed mice at t = 0. (**B–D**) Number of analyzed PSD95 assemblies in EE: 30 min: n = 38, 60 min: n = 68, 120 min: n = 29; Ctr: 30 min: n = 27, 60 min: n = 48, 120 min: n = 38.

The online version of this article includes the following source data for figure 5:

**Source data 1.** Images of all analyzed perforated PSD95 assemblies.

clear trend: the longer the measurement interval, the greater the morphological change (*Figure 5C*); the percentage of PSD95 nanoorganization showing no change decreased from ~40% to ~10%, while the percentage for strong changes nearly tripled for Ctr. For EE housed mice, we observed similar, strong changes within all investigated time intervals; ~20% of the PSD95 assemblies did not change at all and ~80% underwent a change at all time points (*Figure 5B*). Although the percentage changes with respect to t = 0 were similar for different time intervals, it should be noted that this does not exclude dynamic changes between the time points. Thus, changes in Ctr housed mice increased over the observation period and reached a level similar to that of EE housed mice after 120 min. This suggests that the PSD95 nanoorganization is more dynamic in EE mice than in mice raised under

control conditions within the investigated time course. Now, we asked whether the different dynamic in morphological changes would be reflected as well in a different nanoorganization. Therefore, we categorized the nanoorganizations into perforated (a continuous shape with a sub-structure such as ring- or horseshoe-like) or clustered. Fewer nanoorganizations of EE housed mice were perforated and a larger fraction showed three or more clusters compared to Ctr mice (*Figure 5D*). This suggests that the PSD95 nanoorganization might play a substantial role in synaptic remodeling and is specifically shaped by experience or activity.

## Discussion

Interactions with the environment play a key role in restructuring and refining the neuronal circuitry in the brain. We here used time-lapse in vivo STED microscopy, a superresolution light microscopy technique with nanoscale resolution, to study experience-dependent changes of the synaptic nanoorganization as well as its nanoplasticity in the intact brain. We designed a two-color, virtually crosstalk-free in vivo STED microscope with a resolution of 70–80 nm for simultaneous imaging of spine morphology and PSD95 in the visual cortex of anesthetized mice. We found a significantly smaller variability in size of the spine head and PSD95 nanoorganization in EE mice while the average spine head size was increased compared to Ctr mice. Both parameters were highly volatile and we observed an average growth of >20% and shrinkage of >18% for spine heads and PSD95 assemblies within 30 min while percent changes in PSD95 assembly size were slightly larger than those of the spine head. About 3/4 of these changes were positively correlated and ~1/4 showed a negative correlation; EE housed mice exhibited a smaller variance of changes than Ctr mice, and less negatively correlated changes. All parameters exhibited multiplicative downscaling which was significantly different for PSD95 assembly size between EE and Ctr mice. Dynamical rearrangement of the nanostructure was faster in EE housed mice.

### Two-color in vivo STED nanoscopy

Previous studies mainly used two-photon microscopy to study synapse and spine plasticity in vivo and in vitro (*Bosch et al., 2014*; *Cane et al., 2014*; *Meyer et al., 2014*; *Villa et al., 2016*). Due to its limited optical resolution of 300–500 nm, spine brightness is often taken as a measure of size (*Cane et al., 2014*; *Hofer et al., 2009*; *Matsuzaki et al., 2004*). This is not very accurate since the brightness can be influenced by many factors such as expression level of the fluorescence label, the excitation laser intensity, group velocity dispersion, or scattering in the tissue. Therefore, spine brightness is typically normalized to the brightness of the dendrite, making comparative studies and determination of absolute sizes difficult. To circumvent the diffraction limit, two-photon microscopy was combined with STED to achieve nanoscale resolution (*Moneron and Hell, 2009*; *Panatier et al., 2014*). However, a drawback of this approach is the relatively high crosstalk when using two colors. Thus, the most commonly used EGFP and EYFP have a crosstalk of as much as 92% and 27% in the respective other channel, requiring linear unmixing (*Tønnesen et al., 2011*). Unmixing works well for structures which are relatively bright and spatially separated such as for volume labeled spines and astrocytes (*Panatier et al., 2014*) but can produce artifacts in dark and overlapping structures such as the post-synapse and spine we were studying in this project. Therefore we used an approach from two-color STED with organic dyes in the red emission spectrum (*Bottanelli et al., 2016*; *Göttfert et al., 2013*) and designed a two-color STED microscope with two different excitation lasers (one-photon excitation) and two spectrally separated detection channels. This approach gave us an almost negligible crosstalk of 5% for EGFP and 8% for Citrine rendering unmixing obsolete.

### High volatility of spine heads and PSD95 assemblies

We used the transcriptionally regulated antibody-like PSD95.FingR, which efficiently labels endogenous PSD95 (*Cook et al., 2019*; *Gross et al., 2013*) and excludes artifacts that occur after PSD95 overexpression (*El-Husseini et al., 2000*; *Stein et al., 2003*).

Spine heads and PSD95 assemblies were highly volatile already at 30 min intervals. The spine head growth of ~20–30% on average and ~200% maximum (*Figure 3—figure supplement 1*) is very similar to the spine head changes we have reported recently in the cortex of a transgenic mouse over 3–4 days (*Steffens et al., 2021*) and to spine head increase following chemical LTP induction

(*Kopec et al., 2006*; *Otmakhov et al., 2004*) or glutamate uncaging (*Meyer et al., 2014*); thus these changes might reflect potentiation of single spines. The percentage increase in amount of PSD95 after glutamate uncaging reported before, however, was always smaller than the spine volume increase (*Bosch et al., 2014*; *Meyer et al., 2014*); it is therefore interesting that we observe a slightly larger percentage increase of PSD95 assembly area than spine head area at baseline in Ctr and EE housed mice. However, it was shown that MMF anesthesia reduces spiking activity and mildly increases spine turnover in the hippocampus (*Yang et al., 2021*). Thus, the plasticity of spine heads and PSD95 assemblies might be different in the awake state and under intense processing of visual information.

## Temporal changes of PSD95 assemblies and spine heads are largely uncorrelated

While we observed a substantial correlation between the absolute spine head and PSD95 assembly size (r ~0.8, *Figure 2B*), corroborating previous EM studies (*Arellano et al., 2007*; *Harris et al., 1992*), we found only weak correlations between their temporal changes (r ~0.2–0.4, *Figure 3D–G*). This supports in vitro LTP experiments which have shown a temporal delay of ~60 min between spine and PSD95 increase (*Bosch et al., 2014*; *Meyer et al., 2014*). Although these experiments were performed in cell culture and with PSD95 overexpression, they indicate that spine and PSD enlargement might be regulated by different pathways (*Compans et al., 2016*; *Herring and Nicoll, 2016*). However, we found no directional changes that would demonstrate that the PSD95 size systematically follows an increase in spine head size since the cross-correlation is largely symmetric (*Figure 4G*). This could be due to the fact that we only studied baseline changes, without a dedicated stimulation protocol that synchronizes and amplifies the temporal changes.

But how can the time course of AMPA receptor incorporation and PSD95 increase be so different although AMPA receptors are linked to PSD95 by TARPs? A slot model proposes that the actin polymerization during LTP increases the number of slots which can harbor AMPA receptors (*Herring and Nicoll, 2016*). Thus, the increase of AMPA receptors would not depend on an increase of the amount of PSD95 proteins at the synapse but on an activation of binding sites of AMPA receptor to the PSD95 by a still unknown mechanism.

## Synapse and spine sizes of EE housed mice are sharper defined and change concordantly

To study whether different activity conditions during rearing influence synapse and spine size and plasticity, we compared adult mice housed in standard cages with age matched EE housed mice and analyzed synaptic morphology and plasticity in the visual cortex after the critical period. We found a significantly smaller variance and thus narrower distribution of spine head and PSD95 area in EE housed mice. At the moment different models of how neuronal networks adapt to changes in activity to undergo homeostatic plasticity are considered (*Lee and Kirkwood, 2019*). In the model of synaptic scaling, an increase in activity leads to a downscaling of excitatory synapses and vice versa. The downscaling is multiplicative and therefore changes affect the average as well as broadening of the size distribution as shown, for example, in a silenced network (*Hazan and Ziv, 2020*). We did not observe such a stringent synaptic scaling although the variance shows the same tendency; a decrease in PSD95 and spine size variability after enhanced activity of EE housed mice are opposed, and therefore in line with the broadening of the size distribution in silenced networks (*Hazan and Ziv, 2020*) or after deprivation in vivo (*Keck et al., 2013*). This may be supported by recent in vivo observations that in the adult cortex homeostatic plasticity is rather input-specific and not multiplicative for the whole ensemble of synapses (*Barnes et al., 2017*). A sole decrease in variability was described before for the neuronal firing rates after stimulation. This was observed in different brain regions and even when the change in mean firing rate was little (*Churchland et al., 2010*). The authors of this study concluded that the variance decline of the firing rate implies that cortical circuits become more stable. The same might apply to the spines and synapses of EE housed mice for which we observe less extreme values. This may suggest that the synapses are better defined by training in the enriched environment and thus the neural network is more stable. Interestingly, we also observed a tighter correlation between spine head and PSD95 changes for EE housed mice since negatively correlated changes are significantly reduced. As discussed above, however, the dynamic might be influenced by the anesthesia and different in the awake state.

## Stronger multiplicative downscaling in EE housed mice for PSD95

We found that changes of spine heads and PSD95 assemblies correlate with their absolute size for EE and Ctr housed mice; small spines tended to grow and large spines tended to shrink (*Figure 4A–F*). The slope of the linear regression line for such size changes, as plotted in *Figure 4A–F*, can be viewed as a time-dependent multiplicative downscaling factor of a Kesten process (*Ziv and Brenner, 2018*). Statistical models such as the Kesten process offer an explanation on how to link synaptic size fluctuations with the shape and scaling of the synaptic size distribution. For example, it was shown that silencing a neuronal network and thus reducing its activity not only resulted in an average size increase and broader distribution but also in a weaker multiplicative downscaling (*Hazan and Ziv, 2020*). Our observations on PSD95 area changes are in line with these results in such that the increase in activity by enrichment strengthened the multiplicative downscaling over time for PSD95 after 60 and 120 min (*Figure 4F*). Therefore, we find the same tendency in vivo as the in vitro silencing experiment. However, it should be noted that our in vivo measurements were performed under anesthesia and directly after implanting a cranial window; therefore differences are due to the different rearing conditions and not to changes in activity at the moment of the measurement. The in vitro silencing, in contrast, continued over the measurements. The increase in multiplicative downscaling we observe might explain the narrower size distribution of the PSD95 area according to the measurements and simulations by *Hazan and Ziv, 2020*. However, a downscaling factor of 0.7–0.8 which we found for PSD95 in EE housed mice (*Figure 4F*) was observed in vitro by Hazan and Ziv only after ~20 hr. And, this does not explain the decrease in multiplicative downscaling we observed for Ctr housed mice from 30 to 120 min intervals. Moreover, we did not find changes in the Pearsons's correlation coefficient r for the different time intervals such as described by *Hazan and Ziv, 2020*. A refinement of the model and additional measurement will certainly be needed in the future. We note in passing that it should not be surprising that our in vivo experiments do not completely match expectations from artificially silenced neuronal culture experiments. And, as discussed above, there is evidence that homeostatic plasticity is not global but input-specific (*Barnes et al., 2017*).

## Enhanced plasticity of the nanopattern by experience

With our superresolution technique we often observed a perforated nanoorganization of PSD95 undetectable by two-photon or conventional imaging (*Figure 5*). The pattern of the PSD95 nanoorganization labeled with PSD95.FingR was very similar to the structures we have observed earlier in a PSD95-EGFP knock-in mouse with in vivo STED microscopy (*Wegner et al., 2018*). And, this nanopattern is also similar to perforated PSDs reported by electron microscopy (*Arellano et al., 2007*; *Harris and Weinberg, 2012*; *Toni et al., 2001*). It is very difficult to analyze this pattern quantitatively because of its great diversity. As long as the functional consequences are not clear, it is very difficult to design an appropriate analysis routine; for example, should we quantify the area covered with PSD95 only and which role does the distance between clusters or diameter of a ring play? Since we could not apply stringent and justifiable rules for a quantitative analysis, we decided to perform only a visual inspection of the shape, but include all analyzed images into the supplement for transparency. In this way, we found that the nanopattern was different between EE and Ctr and that its structural changes occurred more rapidly in EE housed mice, suggesting greater synaptic flexibility. The functional consequences of this nanopattern, however, are not fully understood. An emerging view is that clusters of pre- and post-synaptic proteins are trans-synaptically aligned in 'nanocolumns' which are organized by activity (*Chen et al., 2018*). Moreover, computational simulations predict that changes in the shape of the PSD are a way to align post-synaptic receptors to pre-synaptic release sites (*Franks et al., 2003*; *Savtchenko and Rusakov, 2014*). This implies that a reorganization of receptors might alter synaptic strength – independently of changes in the amount of receptors (*Chen et al., 2018*). Our finding of an increased structural plasticity of the PSD95 nanoorganization but not increased average size in EE mice might therefore reflect the reported increase of synaptic plasticity after enrichment (*Artola et al., 2006*; *Buschler and Manahan-Vaughan, 2012*).

## Outlook

Our results demonstrate that experience influences the synaptic nanopattern and facilitates structural remodeling of the synaptic nanoorganization. Two-color STED microscopy opens novel avenues for the in vivo investigation of the synaptic nanoorganization and its dynamics in the living brain. In the

**Table 1.** Overview of the primers and endonucleases.
a: produced by PCR, b: generated by hybridization, P: phosphorylated; underlined nucleotides: restriction sites or part of them.

| Target construct | Primer | Restriction sites | DNA-insert |
|---|---|---|---|
| | 5´- agttat<u>gctagc</u>atgggctgtgtgcaatgtaaggataaag aagcaacaaaactgacgatggtgagcaagggcgaggag –3´ | NheI | Myristol (myr)-EGFP[a] |
| | 5´- cgc<u>accggt</u>cttgtacagctcgtccatg-3´ | AgeI | |
| | P-5´-<u>ccggt</u>cggaactggcgcctgaagaatatcaacagc atcaatttcgataaccccgtgtaccagaagaccacagaggat –3´ | AgeI | LDLR(Ct)-part1[b] |
| | P-5´-cagctcatcctctgtggtcttctggtacacgggggttatcgaaa ttgatgctgttgatattcttcaggcgccagttcc<u>ga</u> –3´ | AgeI | LDLR(Ct)-part1[b] |
| | P-5´-gagctgcacatttgcaggtcccaagacgggtacacctatcc aagtcggcagatggtcagcctcgaggacgatgtggcctga<u>gg</u> –3´ | AscI | LDLR(Ct)-part2[b] |
| pAAV-hSyn-DIO-myrEGFP-LDLR(Ct) | P-5´- <u>cgcgcc</u>tcaggccacatcgtcctcgaggctgaccatctgcc gacttggataggtgtacccgtcttgggacctgcaaatgtg –3´ | AscI | LDLR(Ct)-part2[b] |

future this approach could, for example, be extended to study the plasticity of nanocolumns after experience, as well as over longer time intervals in chronic experiments (*Steffens et al., 2021*) and different brain regions.

## Materials and methods
### DNA constructs
A detailed description of the different cloning steps to obtain pAAV-ZFN-hSyn-DIO-PSD95.FingR-Citrine-CCR5TC, with a DIO for the expression of the transcriptionally regulated antibody-like protein PSD95.FingR, can be found in *Willig et al., 2021*.

The plasmid pAAV-hSyn-DIO-myr-EGFP-LDLR(Ct) for dendritic membrane labeling was cloned as follows: First, we PCR amplified a myristoylation (myr) site-attached EGFP including the myristoylation sequence ATGGGCTGTGTGCAATGTAAGGATAAAGAAGCAACAAAACTGACG in the forward primer. Second, we split the C-terminal (Ct) cytoplasmic domains of LDLR (GenBank: AF425607, amino acid residues 813–862) (*Kameda et al., 2008*), finally designated as LDLR(Ct), into two parts (part 1 and part 2), designed 5′ phosphorylated forward and reverse primers for each part and hybridized each pair (*Table 1*). In the third and final step, the endonuclease-digested myr-EGFP PCR together with the LDLR(Ct) part 1 and LDLR(Ct) part 2 were ligated into plasmid pAAV-hSyn-DIO-EYFP digested with AscI and NheI to finally obtain pAAV-hSyn-DIO-myrEGFP-LDLR(Ct).

Generation of the Cre recombinase expression plasmid pAAV-hSyn-Cre was performed as described in *Wegner et al., 2017*.

The crosstalk was determined by expression of a fusion protein consisting of the small peptide Lifeact (LA), which directly binds to F-actin, and a fluorescent protein: pAAV-hSyn-LA-EYFP and pAAV-hSyn-LA-EGFP (*Willig et al., 2021*; *Willig et al., 2014*).

### Virus production
Recombinant AAV particles with mixed serotypes 1 and 2 of the pAAV plasmids encoding the proteins of interest were produced in HEK293-FT cells. The entire procedure is described in detail in *Wegner et al., 2017* and is applied here with the following modifications: After DNaseI treatment (30 min at 37°C), the suspension was centrifuged at 1200 *g* for 10 min. The supernatant was filtered through a 0.45 µm sterile filter (Merck/Millipore, Darmstadt, Germany) and first applied to an Amicon Ultra-15, MWCO 100 kDa, centrifugal filter unit (Merck/Millipore, Darmstadt, Germany), followed by a Vivaspin 500, MWCO 100 kDa, centrifugal concentrator (Sartorius, Göttingen, Germany) to wash the virus in Opti-MEM Medium (ThermoFisher Scientific, Darmstadt, Germany) and concentrate to a final volume of 150 µl.

## Animals

All animal experiments were performed with C57BL/6J female mice reared at the animal facility of the Max Planck Institute for Multidisciplinary Sciences, City Campus, in Göttingen and housed with a 12 hr light/dark cycle, with food and water available ad libitum. Experiments were performed according to the guidelines of the national law regarding animal protection procedures and were approved by the responsible authorities, the Niedersächsisches Landesamt für Verbraucherschutz (LAVES, identification number 33.9-42502-04-14/1463). All efforts were made to avoid animal suffering and to minimize the number of animals used.

## Housing conditions

EE animals were born and raised in the commercially available Marlau cage (Marlau, Viewpoint, Lyon, France) (*Fares et al., 2012*), 580 × 400 × 320 mm$^3$ in size including two floors, providing an extensive exploration area (*Figure 2—figure supplement 1*). Three pregnant female mice were placed in an EE cage about 1 week before delivery. Pups were weaned and split by sex at post-natal day 30. The ground floor of the EE cage consisted of two separate compartments: the smaller part contained food and the larger part contained water, running wheels, and a red house. To get food, the mice had to use a stairway to reach the second floor where a maze was placed. After passing the maze, the mice slide through a tunnel back to the ground floor, directly into the smaller compartment and had free access to food. By passing a one-way door, they could enter the larger part of the ground floor to get water. To increase novelty and maintaining cognitive stimulation, the maze was changed three times per week with a total of 12 different configurations. Ctr mice were born and raised in standard cages of 365 × 207 × 140 mm$^3$ size. They were kept in two to three animals per cage, which were solely equipped with nesting material.

## AAV transduction

Adult (>12 weeks) female C57BL/6J mice were stereotaxically transduced with a mixture of AAV with mixed serotypes 1 and 2. AAV1/2-ZFN-hSyn-DIO-PSD95.FingR-Citrine-CCR5TC, AAV1/2-hSyn-Cre, and AAV1/2-hSyn-DIO-myrEGFP-LDLR(Ct) were stereotaxically injected at the same time into the visual cortex of the left hemisphere at a depth of ~500 µm below the pia transducing mainly layer 5 pyramidal neurons by using the following coordinates: 3.4 mm posterior to the bregma, 2.2 mm lateral to the midline, at an angle of 70° to the vertical, and with a feed forward of ~750 µm. For that purpose mice were anesthetized by intraperitoneal injection of 0.05 mg/kg fentanyl, 5 mg/kg midazolam, and 0.5 mg/kg medetomidin (MMF). The mouse was head fixed with a stereotaxic frame and placed on a heating pad throughout the whole procedure to maintain constant body temperature. The depth of the anesthesia was controlled by monitoring the pulse rate and $O_2$ saturation with a pulse oximeter at the thigh of the mouse and body temperature measured with a rectal temperature probe. A gas mixture with high $O_2$ (47.5 vol%) and $CO_2$ (2.5 vol%) content was applied to the mouse's nose, significantly increasing the oxygenation. The eyes of the mouse were covered with eye ointment and the head was disinfected with 70% ethanol. The skin was cut by an ~0.5 cm long incision and a drop of local anesthetic (0.2 mg mepivacaine) was applied. Then, a 0.5 mm hole was drilled into the skull. 150 nl of AAV containing solution was pressure injected through a glass micropipette attached to a microinjector (Picospritzer III, Parker Hannifin Corp, Cleveland, OH) with an injection rate of ~50 nl/min. After injection, the pipette was kept at the target location for additional 2 min to allow the virus to disperse away. After retracting the micropipette, the incision was closed with a suture. Anesthesia was then antagonized by intraperitoneal administration of 0.1 mg/kg buprenorphine and 2.5 mg/kg atipamezole. The mouse was kept in a separate cage until full recovery and then put back into its original cage, and group-housed in the animal facility until the final experiment. All mice, EE and Ctr housed, were transduced with the same batch of purified virus, that is, the same virus concentration.

## Craniotomy for in vivo STED microscopy

Three to six weeks after viral transduction, a craniotomy was performed as described previously (*Steffens et al., 2020*). In brief, the mouse was anesthetized with MMF, placed on a heating pad and vital functions and depth of anesthesia were controlled throughout the final experiment as described above. The head was mounted in a stereotaxic frame. The scalp was removed and a flat head bar was glued to the right hemisphere to leave enough space for the cranial window above the visual cortex,

rostral to the former viral injection site. After drilling a circular groove (~2–3 mm) into the skull, the bony plate was carefully removed without causing a trauma. The remaining dura and arachnoid mater were carefully removed with a fine forceps. A small tube was positioned at the edge of the opening to drain excess cerebrospinal fluid if necessary. The craniotomy was sealed with a 6 mm diameter cover glass glued to the skull. The mouse was mounted on a tiltable plate which can be aligned perpendicular to the optical axis of the microscope in a quick and easy routine before being placed under the microscope (*Steffens et al., 2020*).

### Culture of primary hippocampal neurons, transduction, and imaging

Primary cultures of rat hippocampal neurons were prepared from P0-P1 Wistar rats (RjHan:WI; The Jackson Laboratory, Bar Harbor, ME) of both sexes according to *D'Este et al., 2017*. Neurons were cultured at 37°C in a humidified atmosphere with 5% $CO_2$ and transduced between 8 and 10 days in vitro with the respective AAVs. After an incubation time of 7 days, live-cell STED imaging was performed at room temperature.

### In vivo two-color STED microscope

A previously described custom-designed STED microscope (*Wegner et al., 2018*; *Willig et al., 2014*) was modified to accommodate virtually crosstalk-free two-color imaging as follows (*Figure 1*). Excitation light (Exc1) provided by a pulsed laser diode emitting blue light at 483 nm (PiLas, Advanced Laser Diode Systems, Berlin, Germany) was complemented by a second excitation beam (Exc2). The beam of a Ti:Sapphire laser (MaiTai; Spectra-Physics, Santa Clara, CA) was split into two, pumping an optical parametric oscillator (OPO; APE, Berlin, Germany) emitting 80 MHz pulses at 595 nm for STED and a supercontinuum device (FemtoWHITE800, NKT photonics, Birkerød, Denmark) generating white light. The white light was spectrally filtered for green light with a bandpass filter (BP1, BrightLine HC 520/5, Semrock, IDEX Health & Science, Rochester, NY) for selective excitation of EYFP or Citrine at 520 nm (Exc2, *Figure 1A and B*). Exc1, Exc2 and the STED beam were co-aligned with dichroic mirrors. After passing a scanning device (Yanus, Till Photonics-FEI, Gräfelfing, Germany) consisting of two galvanometric mirrors for x-y-scanning and relay optics, the three beams were focused by a 1.3 numerical aperture objective lens (PL APO, 63×, glycerol; Leica, Wetzlar, Germany). Additionally, the STED beam was passing a vortex phase plate (VPP; RPC Photonics, Rochester, NY) to create a doughnut-shaped focal intensity pattern featuring zero intensity in the center. Temporal overlap of all three pulsed laser beams was achieved electronically by synchronizing the blue laser diode, Exc1, to the Ti:Sapphire laser and optically by an optical delay line for Exc2. Z-scanning was performed by moving the objective with a piezo (MIPOS 100PL; piezosystem jena GmbH, Jena, Germany). The back-projected fluorescence light was split at 515 nm with a dichroic mirror (DM2, ZT502rdc-UF3; Chroma Technology Corporation, Bellow Falls, VT) into two beams. The shorter wavelength was reflected, filtered by a bandpass filter (BP3, BrightLine HC 504/12; Semrock) and focused onto a multimode fiber for confocal detection connected to an avalanche photodiode (APD, Excelitas, Waltham, MA). The transmitted, longer wavelength fluorescence light was also filtered with a bandpass filter (BP2, H544/23, AHF analysentechnik, Tübingen, Germany) and detected with an APD, respectively.

### In vivo two-color STED imaging of anesthetized mice

The in vivo STED microscopy was performed in layer 1 of the visual cortex at a depth of 5–20 μm below the pia. Spherical aberrations due to the tissue penetration were corrected on the first order by adapting the correction collar of the glycerol immersion objective for the best image quality at each FOV. Both excitation colors were alternated line-by-line; that is, a line was recorded by excitation with blue light (Exc1) and then the same line was recorded again with green excitation light (Exc2). Potential drift between images and movement of the spine or PSD95 was therefore negligible. For imaging we picked different FOV with dendrites parallel to the focal plane. After recording a two-color STED image stack, the position of the motorized micrometer stage (MS-2000, Applied Scientific Instruments, Eugene, OR) was noted and an overview image taken. This process was repeated several times for different positions. After 30 min the micrometer stage was moved back to the first position. The position was confirmed by recording coarse overview images of the dendrite and thereby adjusting the z-position to the right depth. STED images were recorded as z-stacks of ~20–40 μm in x and y with 500 nm axial steps at different positions of the cranial window. All positions were repeatedly imaged

3–4 times at intervals of 30 min to 2 hr. Therefore, some dendritic regions were investigated at the time points 0, 30, and 60 min, while other dendritic region within the cranial window was examined at the time points 0, 60, and 120 min or even at 0, 120, and 240 min. The benefit of a membrane label over the often used volume label is that the brightness of the spines and dendrites is similar whereas with a volume label, spine heads, and dendrites often outshine the small spine neck. Thus, either the much darker neck is not visible between the bright head and dendrite or the detector is saturated at the bright heads and dendrites.

All images were recorded with pixel dwell time of 5 µs, pixel size of 30 × 30 nm$^2$ in x and y, and z-stacks of 500 nm step size. Blue and green excitation power was 4.5 µW, respectively, in the back aperture of the objective. The average STED power in the back aperture was 37–45 mW for static images and 15 mW for time-lapse imaging. EE and Ctr housed mice were imaged in random order but not blindly. Dendrites were randomly picked.

## Data processing and analysis

Confocal and STED images were acquired by the software Imspector (Abberior Instruments, Göttingen, Germany). Size, shape, and brightness analysis was performed manually in Fiji and blind with respect to the housing conditions (*Schindelin et al., 2012*). For spine head and PSD95 analysis, the first and last planes were omitted to ensure that the spine (or PSD95, respectively) was located completely in the focal plane. We only analyzed spines that also bore a PSD and only spines that extended from the dendrite mainly parallel to the focal plane. PSDs directly on the dendrite, most likely representing shaft synapses, and spines pointing upward or downward that could not be clearly resolved were not analyzed. First, images of each channel were processed as follows (Fiji commands shown in capital letters): (1) Smoothing: PROCESS> SMOOTH twice. (2) Brightness adjustment: IMAGE> ADJUST > BRIGHTNESS/CONTRAST set minimum to 1 instead of zero. (3) Overlay both one-color images: IMAGE> COLOR > MERGE CHANNELS. (4) Open ROI manager: ANALYSE > TOOLS > ROI MANAGER. Spine heads and PSD95 assemblies were encircled at their largest extent as shown in *Figure 2A* to compute their area. Perforations or cluster of PSD95 were encircled to include only PSD95. For the analysis of brightness, PSD95 images were processed as described above. Using the ELLIPTICAL SELECTION tool, each PSD95 assembly was encircled and the brightness was displayed by the variable 'RawIntDen', which is the sum of the intensity values of all pixels in the selected area. Spine density was obtained by dividing the total number of spines minus one by the length of the parent dendrite between the first and last spine. The length was measured with the FREEHAND LINE tool, in Z PROJECTION (maximum intensity) images. Branched spines were counted once.

Absolute changes of spine head area or PSD95 assembly area between two time points, t and t + 1, were calculated by $\Delta A_{abs} = A(t + 1) - A(t)$. 'A' denotes the spine head area or the PSD95 area, respectively. Normalized changes of spine head area or PSD95 area were computed by $\Delta A_{norm} = (A(t + 1) - A(t))/(A(t + 1) + A(t))$ and percentage changes by $\Delta A_{\%} = (A(t + 1) - A(t))/A(t)*100\%$.

The PCA was performed with the built-in MATLAB (MathWorks, Natick, MA) function 'princomp'.

Pearson's correlation coefficient and linear regression were computed in GraphPad Prism (version 7.04, GraphPad Software, San Diego, CA). The cross-correlation function was computed for different time intervals $\Delta t$ by

$$CC\left(\Delta t\right) = \frac{\sum_{t,\,n=1}^{N}\left(A1_{\,n}\left(t+\Delta t\right)-\bar{A1}\right)\left(A2_{\,n}\left(t\right)-\bar{A2}\right)}{\sqrt{\sum_{t,\,n=1}^{N}\left(A1_{\,n}\left(t\right)-\bar{A1}\right)^2 \sum_{t,\,n=1}^{N}\left(A2_{\,n}\left(t\right)-\bar{A2}\right)^2}}$$

A1 and A2 denote spine head and PSD95 assembly size, respectively. N stands for the total number of spines, $\Delta t$ the lag time, and $\bar{A}$ the average size of all spine heads, respectively PSD95 assemblies.

## Nanoorganization and morphological changes of PSD95

All non-macular PSD95 nanoorganizations were selected. Their shape was categorized by three blinded scientists into perforated or clustered shapes. Perforated PSDs were of continuous shape but complex such as a ring or horseshoe-like or more twisted. Clustered nanoorganizations were characterized by two or more separated assemblies of PSD95 per spine and the number of clusters was counted. Temporal changes in PSD95 morphology were analyzed as follows. Assemblies which did not

alter their overall morphology or number of clusters were marked as 'no change'. Assemblies showing minor modifications, such as a small movement of a sub-cluster, were classified as 'subtle change'. Changes of the overall morphology such as a smooth continuous PSD falling apart into different clusters were characterized as 'strong change'. All changes were referred to time point t = 0 min.

## Statistical analysis

Statistical analysis was performed in GraphPad Prism. Positively skewed data sets were compared by the Kolmogorov-Smirnov test. Log-transformed data and normalized relative changes were normally distributed and compared by an unpaired t-test with Welch's correction. The number of analyzed mice and spines, p-value, and the specific statistical test performed for each experiment are included in the appropriate figure legend. All tests were applied two-sided where applicable. Probabilities are symbolized by asterisks: $*p < 0.05$; $**p < 0.01$; $***p < 0.001$, $****p < 0.0001$.

## Materials availability

This study did not generate new unique reagents.

## Acknowledgements

We thank Dr Siegrid Löwel (Göttingen University) for suggesting and providing the Marlau cages, the animal facility of the MPI for Multidisciplinary Sciences, City Campus, for excellent support, Dr Karl Deisseroth and Dr Don Arnold for providing plasmids and Jaydev Jethwa for critical reading. This work was funded by the Deutsche Forschungsgemeinschaft (DFG, German Research Foundation) within the DFG Research Center and Cluster of Excellence (EXC 171, Area A1) 'Nanoscale Microscopy and Molecular Physiology of the Brain' (WW, HS, KIW) and under Germany's Excellence Strategy – EXC 2067/1-390729940 (KIW, CG, FW). This work was supported by the Niedersächsisches Vorab through the Göttingen Campus Institute for Dynamics of Biological Networks (FW), and by the Deutsche Forschungsgemeinschaft (DFG, German Research Foundation) through CRC 889, CRC 1286 and the PP 2205 "Evolutionary Optimization of Neuronal Processing" (FW).

## Additional information

### Funding

| Funder | Grant reference number | Author |
|---|---|---|
| Deutsche Forschungsgemeinschaft | EXC171 | Waja Wegner<br>Heinz Steffens<br>Katrin I Willig |
| Deutsche Forschungsgemeinschaft | EXC 2067/1- 390729940 | Carola Gregor<br>Katrin I Willig<br>Fred Wolf |
| Max Planck Institute for Multidisciplinary Sciences | Open Access Funding | Waja Wegner<br>Heinz Steffens<br>Carola Gregor<br>Katrin I Willig |
| VolkswagenStiftung | Göttingen Campus Institute for Dynamics of Biological Networks | Fred Wolf |
| Deutsche Forschungsgemeinschaft | CRC 889 | Fred Wolf |
| Deutsche Forschungsgemeinschaft | CRC 1286 | Fred Wolf |

The funders had no role in study design, data collection and interpretation, or the decision to submit the work for publication.

## Author contributions
Waja Wegner, Conceptualization, Formal analysis, Investigation, Methodology, Resources, Visualization, Writing – original draft, Writing – review and editing; Heinz Steffens, Investigation, Methodology, Writing – review and editing; Carola Gregor, Resources, Writing – original draft, Writing – review and editing; Fred Wolf, Formal analysis, Methodology, Writing – original draft; Katrin I Willig, Conceptualization, Data curation, Formal analysis, Funding acquisition, Methodology, Project administration, Supervision, Validation, Visualization, Writing – original draft, Writing – review and editing

## Author ORCIDs
Katrin I Willig http://orcid.org/0000-0002-1860-334X

## Ethics
Experiments were performed according to the guidelines of the national law regarding animal protection procedures and were approved by the responsible authorities, the Niedersächsisches Landesamt für Verbraucherschutz (LAVES, identification number 33.9-42502-04-14/1463). All surgery and imaging was performed under anesthesia, and all efforts were made to minimize animal suffering and the number of animals used.

## Decision letter and Author response
Decision letter https://doi.org/10.7554/eLife.73603.sa1
Author response https://doi.org/10.7554/eLife.73603.sa2

# Additional files

## Supplementary files
• Transparent reporting form

## Data availability
Source data files of all analysed data are included in the submission.

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
