## [Editor Report]

Synapses mediate information transmission in the brain, and part of the synaptic structure called spines are the receiving end of signal transfer between neurons. Using a custom-built superresolution microscope, the study reveals the nanoscale structural dynamics of individual spine shape and its resident scaffolding protein PSD95 simultaneously, in mouse cortex in vivo. Aspects of the structural dynamics are found to differ depending on whether mice have been reared in a simple housing or in an enriched environment, the latter condition being associated with enhanced activity.

---

## [Decision Letter]

**Decision letter after peer review:**

Thank you for submitting your article "Environmental enrichment enhances patterning and remodeling of synaptic nanoarchitecture revealed by STED nanoscopy" for consideration by *eLife*. Your article has been reviewed by 3 peer reviewers, one of whom is a member of our Board of Reviewing Editors, and the evaluation has been overseen by Lu Chen as the Senior Editor. The reviewers have opted to remain anonymous.

The two essential revisions concern (1) clarifying the biological question being addressed and (2) validating the use of intrabody to follow the nano-structure of PSD95. The full reviews are appended below, which will help clarify the concerns of the individual reviewers, including the two points. In addition, the authors should address all the points raised in the individual reviews. The requested revision does not involve new experiments, and require mostly re-analysis of data and re-writing with clarifications and additional explanations.

*Reviewer #1 (Recommendations for the authors):*

This study capitalizes on the crosstalk-free two-color STED developed by the authors (Willg et al., Cell Rep 2021) to examine the dynamic changes in synapse structure in mouse visual cortex. Specifically, imaging the superficial dendrites of layer V pyramidal neurons, the authors compared how rearing mice in enriched environment (EE) affects spine morphology and the dynamics of PSD95 scaffolding protein within spines, compared to mice reared in control housing. Curiously, EE mice show less variable spine head volumes and PSD95 areas compared to control mice, while the spine head volume is larger but not PSD95 area in EE group compared to the control group. Moreover, nano-organization of PSD95 displays more prominent changes in EE compared to controls. The authors provide data of excellent quality, and the properties of subtle differences in the dynamics of spine head size and PSD95 organization would be of interest to cellular neuroscientists. Nevertheless, there is a large gap between spine structure dynamics and EE rearing. The causal relationship between the changes in PSD95 and spine size and the presumed enhanced sensory stimulation in EE should be better defined, or at least framed in a way that the results could be better interpreted. Moreover, given that the intrabody method to label PSD95 has not been used to examine nano-organization in detail, one should first validate the utility of the approach by establishing that it provides an accurate readout of the endogenous protein, for example, by using fluorescence immunolabelling to compare PSD95 and FingR signals.

1) That PSD95 intrabody has little effect on synapse organization has been examined at the level of conventional light microscopy. Given that the intrabody tagged with a fluorescent protein is of considerable size (~37 kD, according to Gross et al., 2013), could the authors exclude the possibility that there is no potential artefacts from steric hinderance, for instance?

2) To what extent does the amount of nanobody expressed per cell affect the dynamic behavior of PSD95? Could one be certain that there is no apparent relationship between the level of PSD95 intrabody expression per cell and PSD95 dynamics? In Figure S3, it would be more informative to compare the relationship between the brightness of PSD95 signal and the size of spines, the latter being measured by an independent signal. If the level of expression of PSD95 intrabody has no effect on PSD95 dynamics, then one would not expect to see a relationship, and this should be tested.

3) How does the PSD95 area measured using the intrabody compare to previous data obtained from PSD95-EGFP knock-in mice as reported in Wegner et al., 2018?

4) Figure 2C-J. Reduced variance in the sizes of spine head and PSD95 in the EE group seems to be due to the loss of smaller spine heads and PSD95 areas in the group. If one excludes the smallest spine heads and PSD95 areas, then is there any difference in the distribution?

5) Line 223. "However, whether dynamic changes between these two features are also strongly linked has remained unknown." Contrary to the statement, temporal uncoupling of spine head size changes in PSD95 increase has been noted previously (cf. Lines 48-53).

6) Figure 3G-H. The rationale for assessing the total variance of PSD95 area and spine size combined, for the comparisons between control and EE is not clear. Also, why does the variance for control show a peak at 60 min but decline at 120 min?

7) Figure 4F. As with the comment above, it is not clear why the slope of PSD95 area change should plateau at 60 min and show little increase at 120 min relative to 60 min for continuous baseline imaging.

*Reviewer #2 (Recommendations for the authors):*

1. The method is a nice advance that will be important for the field.

2. The motivation for the biological part of the study is lacking. The overarching question is not clear to me. One smaller question the authors are asking is if the correlation between PSD95 and spine head size is maintained in a short time window of plasticity. I am not sure why that is an important question. They also find that EE reduces variation in spine head size, but it is not clear the biological importance or consequences of a smaller variation in spine head size. Why is this an important analysis to do? The same can be said (noted below) about changes in PSD morphology. Throughout the paper, the authors should have the motivation for each analysis and link it back to their main overarching question. It is hard to say whether this study is an important biological advance because I am not sure what question they are really trying to address.

3. The introduction is really difficult to follow and reads a bit like a stream of consciousness. Please break it into paragraphs with themes. The introduction does not set up an overarching biological question and it should. Why have the authors done these particular experiments and analyses?

4. In a number of places in the paper, the language is difficult to read and at times overly complicated in structure. The authors often make style choices to not use commas surrounding explanatory clauses, but including commas would help with parsing many of the sentences. There are many typos throughout the manuscript, particularly with prepositions. A strong edit to make the language clear and direct would be very helpful for readers.

5. In line 84-85, the authors say that the dynamics of individual synapses in enriched environment are unknown. This is not entirely true. Yang et al., 2009 specifically looked at spine dynamics in vivo with enriched environment (PMID: 19946265), which should be cited here. Greifzu et al., 2014 also examines this general question in visual cortex by looking at E/I balance (and thus indirectly synapses) in enriched environments (PMID: 24395770). This study should also be cited.

6. In line 106-107, the authors say that 'Previous attempts featuring STED microscopy of EGFP and EYFP by two-color detection were suffering of high crosstalk requiring channel unmixing.' Could the authors please say what the issues were previously and what they have done to solve that problem? It is not clear to me, but the explanation would help highlight their methodological development.

7. As mentioned in the section above, I cannot find how long the authors waited after the cranial window surgery until they imaged, but if it is less than four weeks, they need to comment on the effects of inflammation on their synaptic results. This is critical for the interpretation.

8. In lines 287-289, the authors state that bigger spines tend to get smaller and smaller spines tend to get bigger. Given that there is a limit on spine head size, I think that the default hypothesis would be that this reflects regression to the mean. I am not sure why the authors have included this analysis, but they should either show controls that indicate it is not regression to the mean or remove this analysis from the manuscript.

9. As stated in the section above, it is not clear to me the biological relevance of changes in nanoorganization of PSD95. What are the biological consequences or significance of a shift in the nanoorganization for the function of the synapse? Also, could this analysis be quantiative, rather than just descriptive?

10. Lines 346-347, what does subtle change or strong change mean for a PSD95 morphology? Can this be quantified as a percentage change of some type? Could the authors also please explain the biological significance or consequences of this change?

11. In lines 457-458 and 472-473, the authors should cite the original paper that showed that in vivo scaling is input specific, Barnes et al. 2017 (PMID 24395770), not the review that they have cited here.

*Reviewer #3 (Recommendations for the authors):*

1) The data quality is amazing, it is very impressive that this resolution is possible in a breathing animal with a beating heart, using relatively slow scanning microscopy. You should mention the complex procedure you developed to ensure stability, normal orientation and biocompatible surface of the cranial window, pointing to the Methods paper for detail. You have earned your bragging rights.

2) What kind of anesthesia was used during imaging? How much time elapsed between the onset of anesthesia and the first imaging time point (t = 0 min)? Were the imaging experiments performed blind with respect to the housing conditions?

3) lines 235-238: "This means..." This sentence is confusing, delete. The correlation is clearly described in the next sentence. Same for the figure title: "positively and negatively correlated" - I see only weak positive correlations on the population level.

In general, Fig. 3 is a bit confusing due to the separate analysis of 3 time points, but no discussion about what happened at t = 0 (onset of anesthesia?). Therefore, the reader is left wondering if the fact that correlation and variance are more or less tight at different time points carries any biological relevance, or if these are supposed to be repeated measures of a Kesten process at work, or perhaps a control for stable imaging conditions? If this is about detecting differences between EE and control, wouldn't it make more sense to pool all time points?

4) Fig 4F, control animals: I have a hard time understanding how there can be shrinkage at 30 min sampling intervals, but not at longer intervals. Does the dt30min group only contain (30 min - 0 min), or also (60 min - 30 min)? Does this mean initially shrunken PSDs grow back again? Has this something to do with the onset of anesthesia? Please explain/interpret this result.

5) Fig. 4G: This analysis is great, but its significance might be difficult to understand for some readers. It might be worth pointing out that if there were a temporal sequence, e.g. first spine size expansion, then PSD enlargement, this would result in little correlation when comparing just two time points. Thus, the cross-correlation analysis. One could even do a little simulation to illustrate how the cross correlation would look like if changes were linked with a delay (this is optional).

6) Did some spines disappear completely during the period of observation?

[Editors' note: further revisions were suggested prior to acceptance, as described below.]

Thank you for resubmitting your work entitled **"**Environmental enrichment enhances patterning and remodeling of synaptic nanoarchitecture as revealed by STED nanoscopy**"** for further consideration by *eLife*. Your revised article has been evaluated by Lu Chen (Senior Editor) and a Reviewing Editor.

The manuscript has been significantly improved but there are some remaining issues, mostly concerning aspects of the biological context of the study, that need to be addressed, as outlined below:

Line 32-33: "… synaptic strength is set more precisely."

The term "precise" implies that there is a specific value around which the synaptic strength is set, which is not clear in this case. It is suggested that the authors use a more neutral expression such as ".. synaptic strength is set more uniformly" to describe the key observation.

Line 62: "… which was confirmed recently (Holler et al., 2021)."

It would be informative to indicate that the study involved EM (CLEM) analysis, and should be rephrased, for example, "… which was confirmed recently at ultrastructural resolution..".

Line 67: "… directly after LTP."

->.. directly after LTP induction.

Line 70: "… with a delay of ~1 hour after LTP"

As with above, potentiation itself can last for hours, and thus the timeframe being referred to needs to be clear.

-> e.g. … with a delay of ~1 hour after inducing LTP

Line 76: "… of the spine and postsynapse at increased activity and baseline in vivo."

The increased activity is an assumption, and this should be reflected in the statement. For example, one could rephrase along the lines of "… of the spine and postsynapse in vivo using mice reared in enriched environment representing increased activity conditions and normal housing representing baseline conditions."

Line 123

Is it meant that in one possibility, EE could directly affect "pre-existing" synapses rather than "all" synapses?

Line 190

In addition to the reference to Steffens et al., 2020, it would be helpful to the reader if the authors briefly mention what particular procedural features help ensure the craniotomy be atraumatic as possible.

Line 580: "To study activity-dependent changes, we compared adult mice housed in standard cages with age matched EE housed mice and analyzed synaptic morphology and plasticity …."

The motivation for the study is somewhat misguided and unclear, since the present work assessed the influence of activity by subjecting mice to activity-enhanced conditions by means of EE, and did not directly examine activity-dependent changes per se. Moreover, as acknowledged by the authors, there may be an issue also of anesthesia. The starting sentence as well as the rest of the discussion in the paragraph need to reflect these points.

Paragraph starting from Line 606

As with the comment above, the caveats of the present experimental design, including the effects of anesthesia, should be acknowledged.

---

## [Author Response]

The two essential revisions concern (1) clarifying the biological question being addressed and (2) validating the use of intrabody to follow the nano-structure of PSD95. The full reviews are appended below, which will help clarify the concerns of the individual reviewers, including the two points. In addition, the authors should address all the points raised in the individual reviews. The requested revision does not involve new experiments, and require mostly re-analysis of data and re-writing with clarifications and additional explanations.

We would like to thank all reviewers for acknowledging our work and for their thoughtful comments and constructive criticism. The remarks were very helpful and improved the manuscript. We have addressed all points raised below and hope the reviewers will find our revised version suitable for publication.

Reviewer #1 (Recommendations for the authors):This study capitalizes on the crosstalk-free two-color STED developed by the authors (Willg et al., Cell Rep 2021) to examine the dynamic changes in synapse structure in mouse visual cortex. Specifically, imaging the superficial dendrites of layer V pyramidal neurons, the authors compared how rearing mice in enriched environment (EE) affects spine morphology and the dynamics of PSD95 scaffolding protein within spines, compared to mice reared in control housing. Curiously, EE mice show less variable spine head volumes and PSD95 areas compared to control mice, while the spine head volume is larger but not PSD95 area in EE group compared to the control group. Moreover, nano-organization of PSD95 displays more prominent changes in EE compared to controls. The authors provide data of excellent quality, and the properties of subtle differences in the dynamics of spine head size and PSD95 organization would be of interest to cellular neuroscientists. Nevertheless, there is a large gap between spine structure dynamics and EE rearing. The causal relationship between the changes in PSD95 and spine size and the presumed enhanced sensory stimulation in EE should be better defined, or at least framed in a way that the results could be better interpreted. Moreover, given that the intrabody method to label PSD95 has not been used to examine nano-organization in detail, one should first validate the utility of the approach by establishing that it provides an accurate readout of the endogenous protein, for example, by using fluorescence immunolabelling to compare PSD95 and FingR signals.

We thank the reviewer for the valuable comments. We have refined the introduction and explained our motivation in more detail. We also have justified the accuracy of the PSD.FingR label. Please find the details below.

1) That PSD95 intrabody has little effect on synapse organization has been examined at the level of conventional light microscopy. Given that the intrabody tagged with a fluorescent protein is of considerable size (~37 kD, according to Gross et al., 2013), could the authors exclude the possibility that there is no potential artefacts from steric hinderance, for instance?

This aspect is indeed vital to our overall approach. We absolutely agree that steric hindrance, or labelling density in general is a major problem that is particularly evident in superresolution microscopy. Gross et al., 2013 show nicely that “the expression patterns of their endogenous target proteins or the number or strength of synapses” was not affected by PSD95.FingR. It is very difficult to prove whether there are potential artefacts from steric hindrance, which were not recognized by this original publication. However, we think that PSD95.FingR excellently recognizes the PSD95 nanopattern without artefacts for the following reasons. First, we double labeled PSD95 with another intrabody, PF11 which shows a very high overlap with PSD95.FingR (Author response image 1). Second, the PSD95.FingR labeled nanostructure of endogenous PSD95 is very similar to our previously recorded in vivo STED data of a PSD95-EGFP knock-in mouse (Author response image 2). Third, the nanopattern of PSD95, which we observe is very similar to the perforated post-synaptic density of electron microscopy images (Arellano et al., 2007; Harris and Weinberg, 2012; Toni et al., 2001). Forth, FingR.PSD95 is less than half the size of PSD95 (cf. Figure 2G, (Gross et al., 2013)). Together, this indicates that PSD95.FingR is an excellent label of PSD95.

**Author response image 1. sa2fig1:** Double labelling of PSD95 with two different intrabodies, PSD95.Fing and PF11. PF11 recognizes palmitoylated PSD95 (Fukata et al., 2013) and was cloned with the hSyn promoter and the orthogonal transcriptional regulation IL2RGTC (Gross et al., 2013) in the same pAAV backbone as PSD95.FingR. PF11 fused to EGFP and PSD95.FIngR fused to Citrine were co-expressed in cultured hippocampal neurons as described in (Wegner et al., 2017) and imaged with our two-color setup in confocal mode. Both labels show a very high degree of co-localization. Of note, in our hands the expression level of PF11 was not bright enough to record superresolution STED images but PSD95 nanodomains with PF11 labeling were shown in (Fukata et al., 2013). PF11 was a gift from Masaki Fukata, National Institute for Physiological Sciences, Japan.

**Author response image 2. sa2fig2:** Side-by-side comparison of two different labelling schemata of PSD95 imaged by in vivo STED microscopy. (**A**) Selection of PSD95 assemblies showing a nanopattern of PSD95-EGFP knock-in mouse as published in the supplementary material of (Wegner et al., 2018). (B) PSD95.FingR-Citrine (green) and membrane label (magenta) which were analyzed for Figure 5, this manuscript (Images are included in Figure 5–source data 1). (**A**, **B**) All images were recorded in the visual cortex of an anaesthetized mouse. The PSD95 nanopattern of the EGFP knock-in mouse (**A**) and viral expression of PSD95.FingR-Citrine (**B**) are very similar.

We added the following to the discussion: “The pattern of the PSD95 nanoorganization labeled with PSD95.FingR was very similar to the structures we have observed earlier in a PSD95-EGFP knock-in mouse with in vivo STED microscopy (Wegner et al., 2018). This nanopattern is also similar to perforated postsynaptic densities reported by electron microscopy (Arellano et al., 2007; Harris and Weinberg, 2012; Toni et al., 2001).”

2) To what extent does the amount of nanobody expressed per cell affect the dynamic behavior of PSD95? Could one be certain that there is no apparent relationship between the level of PSD95 intrabody expression per cell and PSD95 dynamics? In Figure S3, it would be more informative to compare the relationship between the brightness of PSD95 signal and the size of spines, the latter being measured by an independent signal. If the level of expression of PSD95 intrabody has no effect on PSD95 dynamics, then one would not expect to see a relationship, and this should be tested.

We thank the reviewer for this valuable comment. It is a very difficult endeavor to conscientiously prove that the amount of PSD95.FingR expression has no effect on the PSD95 dynamics. However, for the following reasons we think that this aspect does not influence our results. First, due to the transcriptional regulation system we employ for the PSD95.FingR expression (see Author response image 3 for explanation) we did not detect a significant variation in PSD95 brightness between cells, which is indeed a common problem with conventional overexpression of fusion proteins.

We explain this system now in the Result section.” Second, the morphological changes of the PSD95 nanopattern were analyzed in the same way as in our previous publication of the PSD95-EGFP knock-in mouse (cf. Figure 2E, (Wegner et al., 2018)) and we found similar temporal changes for our Ctr mice. Third, we used the same batch of purified virus for all experiments shown, i.e. the same virus concentration; EE and Ctr mice were labeled by exactly the same protocol.”

We added this information to the Methods. “Fourth, we studied temporal changes on the time scale of 30 to 120 minutes which is relatively slow, much slower than a free diffusion which would be on a time scale below one second. Thus, we do not expect an influence of the attached label on such a slow nanoplasticity.”

Unfortunately, we did not measure the spine size by another, independent signal. In addition, as shown in Figure 2B, spine head area and PSD95 area only show a correlation coefficient of ~ 0.8 and thus such a correlation as suggested by the reviewer would be rather coarse.

**Author response image 3. sa2fig3:** Transcriptional regulation of PSD95.FingR. (**A**) PSD95.FingR expression is controlled by a negative feedback regulation so that once endogenous PSD95 binding sites are saturated, unbound PSD95.FingR moves to the nucleus due to a nuclear localization sequence that is part of the CCR5 zinc finger domain. Binding of the repressor KRAB-A to the promoter via the zinc finger inhibits further transcription and expression of PSD95.FingR. Thus we expect a similar brightness of the PSD95.FingR label when in saturation. Adapted from Figure 3G in (Gross et al., 2013). (**B**) After an in vivo experiment the mouse was perfused and the brain was sliced. A confocal image of layer 5 cortex shows a bright Citrine label of unbound PSD95.FingR in the nucleus indicating that the PSD95 binding sites in this pyramidal neuron were saturated. Since all measurements were performed at the same conditions we expect a similar brightness of PSD95 in all images.

3) How does the PSD95 area measured using the intrabody compare to previous data obtained from PSD95-EGFP knock-in mice as reported in Wegner et al., 2018?

Unfortunately, the sizes of these two datasets cannot be compared in absolute terms for the following reasons. In the knock-in mouse, PSD95 is ubiquitously labeled, i.e. all cell types which express PSD95 will be fluorescently tagged. In layer 1 this includes apical dendrites of layer 5 and layer 2/3 pyramidal neurons as well as interneurons. The PSD95.FingR encoding AAV, however, was injected into layer 5 and thus we imaged mainly layer 5 apical dendrites in layer 1. Spine head and PSD sizes are highly variable within one cell, but averages of these values also change between cell type and brain area. For example, spine heads of layer 5 are larger than those of layer 2/3 (Konur et al., 2003) and the PSD area average size is different between brain regions (Table 1, (Harris and Weinberg, 2012)). Therefore, sizes of the layer 5 PSD95 labeled with FingR cannot be compared with the ubiquitous PSD95 knock-in label.

Another difficulty in comparing these datasets is the different method how the PSD95 area was analyzed. In Wegner et al. we measured only the length and width (width was measured only for some assemblies) and averaged these values. For the PSD95.FingR we refined this method and encircled now each PSD95 assembly, which is more accurate especially for the complex PSD95 shapes. Thus, we determined an average diameter in Wegner et al. 2018, but an average area for the PSD95.FingR. However, in a first draft of the current manuscript we have analyzed the length and width of the PSD95 assemblies similarly to Wegner et al. 2018. For comparison, we show a histogram of this early length analysis in Author response image 4. It shows that the PSD95 assemblies of layer 5 apical dendrites are larger than that of the PSD95 knock-in mouse which most likely reflects the large spine heads of layer 5 neurons (Konur et al., 2003) as discussed above.

**Author response image 4. sa2fig4:** PSD95 assembly size from in vivo STED measurements in layer 1 mouse visual cortex for different cell types. (**A**) Size/length measurement of ubiquitous PSD95 assemblies of a PSD95-EGFP knock-in mouse as published in Wegner et al., 2018. (**B**) Average of length and width of PSD95 assemblies of layer 5 pyramidal neurons labeled with PSD95.FingR. Data set Ctr mice, first time point.

To address this difference, we added to the Result section: “This size of the PSD95 area of layer 5 pyramidal neurons is slightly larger than our previously reported diameter of 354 nm which corresponds to ~ 0.10 µm^2^ for a circular distribution, obtained in a ubiquitously expressing PSD95-EGFP knock-in mouse (Wegner et al., 2018); therefore, the larger size of the PSD95 area could reflect the larger size of the spine heads of layer 5 pyramidal neurons (Konur et al., 2003).”

4) Figure 2C-J. Reduced variance in the sizes of spine head and PSD95 in the EE group seems to be due to the loss of smaller spine heads and PSD95 areas in the group. If one excludes the smallest spine heads and PSD95 areas, then is there any difference in the distribution?

A reduction in variance means that there are fewer extreme values. The reduction in variance comes along with an increase in spine head size for EE mice which indicates indeed a loss of smaller spine heads. We agree with the reviewer on this point. However, this can already be seen well in Figure 2E; therefore we prefer to show only the entire distribution. Since there is no definition on what is a small spine, a cut-off would be rather arbitrary and change the shape of the distribution.

We included this observation into the Result section: This is manifested by a significantly larger spine head area (Figure 2G) and smaller variance of the size distribution (Figure 2I) of EE housed mice, which might imply a preferential loss or adaptation of small spines (Figure 2E).

5) Line 223. "However, whether dynamic changes between these two features are also strongly linked has remained unknown." Contrary to the statement, temporal uncoupling of spine head size changes in PSD95 increase has been noted previously (cf. Lines 48-53).

Thank you for this comment. This sentence now reads: However, on which time scale dynamic changes between these two features are linked in vivo has remained unknown.

6) Figure 3G-H. The rationale for assessing the total variance of PSD95 area and spine size combined, for the comparisons between control and EE is not clear. Also, why does the variance for control show a peak at 60 min but decline at 120 min?

The total variance is a measure for the overall variability of the two parameters, spine head size and PSD95 area; it shows that these parameters are more variable in Ctr than EE housed mice. However, since the total variance is just the sum of the variances for PC1 and PC2 this information is redundant and we moved the total variance to the extended figure. Instead, we took up comment “minor point 3” of reviewer 3 and summed up the different time intervals to more clearly show the difference between EE and Ctr housed mice. While the variance is lower for both, PC1 and PC2 in EE housed mice this decrease is highly significant only for PC2, which indicates a strong decrease in negatively correlated changes. Cf. also comment to reviewer 3, added panel Figure 3G, J.

The peak of the variance at 60 min and decline at 120 min for Ctr is not statistically significant and thus this peak might be just a statistical fluctuation. For a conservative interpretation of the data, we only claim that the variance of PC1 and PC2 is larger for Ctr at all time intervals and anti-correlated changes represented by PC2 are significantly increased for Ctr mice (Figure 3J)

7) Figure 4F. As with the comment above, it is not clear why the slope of PSD95 area change should plateau at 60 min and show little increase at 120 min relative to 60 min for continuous baseline imaging.

We agree with the reviewer that the reason for the changes of the slopes is not clear; in particular, it is not clear why the slope increases for Ctr. Our measurements are to our knowledge the first showing such fluctuations in size in vivo and simulations of these temporal changes are difficult since the mechanism which drives these changes is not known in detail. However, the overall differences between EE and Ctr fit very well with what was observed for the synaptic size and fluctuation parameters in a silenced network (Hazan and Ziv, 2020). Hazan and Ziv found that synaptic silencing increased the synaptic size distribution in mean size and width, as well as weakened the multiplicative descaling. Our model of increased activity by EE leads to an opposing effect; the synaptic size distribution is narrower and the multiplicative descaling increases, i.e. smaller slope for PSD95 at 60 min and 120 min intervals which could explain their narrower size distribution. Future studies will follow to examine these size changes also at longer time intervals, for example by combining with our recently established chronic imaging method (Steffens et al., 2021)

We have refined the discussion on this topic in Discussion section “Stronger multiplicative downscaling in EE housed mice for PSD95”

Reviewer #2 (Recommendations for the authors):1. The method is a nice advance that will be important for the field.

We thank the reviewer for acknowledging our progress on two-color in vivo STED microscopy.

2. The motivation for the biological part of the study is lacking. The overarching question is not clear to me. One smaller question the authors are asking is if the correlation between PSD95 and spine head size is maintained in a short time window of plasticity. I am not sure why that is an important question. They also find that EE reduces variation in spine head size, but it is not clear the biological importance or consequences of a smaller variation in spine head size. Why is this an important analysis to do? The same can be said (noted below) about changes in PSD morphology. Throughout the paper, the authors should have the motivation for each analysis and link it back to their main overarching question. It is hard to say whether this study is an important biological advance because I am not sure what question they are really trying to address.

We regret that our introduction and motivation was not clear enough. However, we thank the reviewer for asking these critical questions which have helped us to restructure the introduction and to include new aspects to the discussion. Our response point-by-point:

- Overarching question?

There is general consensus that acquisition of memory activates or forms a specific assembly of synapses. Thus, spines emerge, disappear, or change with cellular processes underlying learning, and even “remember” previous sensory experience (Poo et al., 2016). It is also clear that learning induces structural and functional synaptic changes similar to long-term potentiation (LTP) protocols. In this concept of learning, however, the maintenance of memory critically depends on the stability of the underlying synaptic connections. However, there is substantial evidence that synaptic structures are highly volatile intrinsically as such that synaptic connections undergo continuous spontaneous remodeling without any activity. Previous in vivo studies have focused primarily on the persistency of spines and synapses in terms of their appearance and elimination and estimated the spine size from fluorescence intensity measurements; directly assessed changes in synapse or spine head size and visualizations of the synaptic nanoorganization in vivo are missing. With our approach of superresolution two-colour in vivo STED microscopy we address with nanoscale resolution (1) the plasticity of spine heads and synapses at baseline at time scales similar to LTP processes; (2) the correlation between PSD95 and spine head size changes; (3) the plasticity of the PSD95 nanoorganization; and (4) whether enhanced activity changes the structure and/or plasticity of these measures.

- Why is the correlation between PSD95 and spine head size important?

in vitro experiments have shown a temporal detuning of PSD95 and spine head size after glutamate uncaging (Meyer et al., 2014). We set out to determine whether such a temporal shift also occurs at baseline in vivo for two reasons: In the literature both parameters, spine heads and PSD95, are often used as a measure for the synaptic strength since their size can be determined relatively easily and repetitively in vivo for large sets of data. Strongly uncorrelated changes would raise the question of which parameter is the better correlate for synaptic strength. Secondly, AMPAR mediated currents were shown to increase simultaneously with the spine head directly after LTP. Given that AMPARs are anchored to the synapse at PSD95 via TARP, a temporal decorrelation between spine head and PSD95 accumulation size suggests that the number of PSD95 is not the rate limiting parameter for (short-term) synaptic plasticity; thus PSD95 might provide slots for AMPAR which are activated by an unknown mechanism on stimulation.

- Smaller variation in spine head/PSD95 area? Biological consequence?

The most widely studied form of activity dependent synaptic structural changes is chronic silencing or deprivation of synaptic activity which is compensated by a multiplicative increase in the strength of excitatory synapses, characterized by an increase in average synapse or spine size and broadening of the distribution. As such, for example, described for deprivation in vivo (Keck et al., 2013) or silenced cortical neuronal cultures (Hazan and Ziv, 2020). Interestingly, we did not find such a multiplicative scaling of synaptic or spine head sizes when comparing enhanced-activity EE and Ctr housed mice.

However, a decrease in variability was described before for the neuronal firing rates after stimulation. This was observed in different brain regions and even when the change in mean firing rate was little (Churchland et al., 2010). The authors of this study conclude that the variance decline of the firing rate implies that cortical circuits become more stable. The same might apply to our observation of smaller variance of synapse sizes; we observe less extreme values, a smaller variation of size changes and less negatively correlated changes for EE housed mice which might suggest that sizes are better defined. Thus, the neuronal network might be of higher stability after training in the enriched environment.

Further experiments are certainly needed to clarify the biological impact of these findings; for example, how do spine heads and PSD95 change in other brain regions such as in the motor cortex or in other cortical layers? How much does the EE change the activity in the visual cortex? Which implication does the variance of these sizes have on models of synaptic stability?

- Why are changes in PSD morphology important?

Perforated PSD95 nanoorganizations as well as AMPA receptors are found mainly on large spines. AMPAR are anchored to PSD95 via stargazin/TARP. Simulations have shown that AMPAR current amplitude drops significantly already at 50 nm offset between presynaptic glutamate release side and AMPA cluster (Haas et al., 2018). Changes in the PSD95 nanoorganization might therefore be a fast mechanism to align postsynaptic receptor to presynaptic release sites.

We have restructured the introduction and complemented the discussion to include these valuable points

3. The introduction is really difficult to follow and reads a bit like a stream of consciousness. Please break it into paragraphs with themes. The introduction does not set up an overarching biological question and it should. Why have the authors done these particular experiments and analyses?

We have revised the introduction and subdivided into paragraphs with themes as suggested.

4. In a number of places in the paper, the language is difficult to read and at times overly complicated in structure. The authors often make style choices to not use commas surrounding explanatory clauses, but including commas would help with parsing many of the sentences. There are many typos throughout the manuscript, particularly with prepositions. A strong edit to make the language clear and direct would be very helpful for readers.

We have revised the language.

5. In line 84-85, the authors say that the dynamics of individual synapses in enriched environment are unknown. This is not entirely true. Yang et al., 2009 specifically looked at spine dynamics in vivo with enriched environment (PMID: 19946265), which should be cited here. Greifzu et al., 2014 also examines this general question in visual cortex by looking at E/I balance (and thus indirectly synapses) in enriched environments (PMID: 24395770). This study should also be cited.

We thank the reviewer for pointing out these references which we included it in the introduction of structural changes and synaptic plasticity by EE.

However, Yang et al. focused on the spine turnover and we refer in our manuscript with “dynamics of individual synapses” to the spine and synapse substructure of the whole ensemble of spines, including persistent spines. According to Yang et al. and others, a large fraction of the spines persists throughout life. We hope we could clarify that aspect in the introduction.

6. In line 106-107, the authors say that 'Previous attempts featuring STED microscopy of EGFP and EYFP by two-color detection were suffering of high crosstalk requiring channel unmixing.' Could the authors please say what the issues were previously and what they have done to solve that problem? It is not clear to me, but the explanation would help highlight their methodological development.

We added to the results: “The challenge for in vivo two-color STED microscopy is to find an in vivo compatible pair of fluorescent molecules with similar emission spectra so that it can be depleted with the same STED beam, but at the same time can be temporally or spectrally separated (Willig et al., 2021). Previous attempts featuring STED microscopy of EGFP and EYFP utilized a single excitation wavelength and two-color detection, which suffered from high crosstalk, and therefore required a linear unmixing of channels (Tønnesen et al., 2011). Channel unmixing, however, requires large signal to noise levels. To reduce crosstalk and thus avoid the necessity of channel unmixing, we extended our previously described in vivo STED microscope (Willig et al., 2014) by an additional two-color excitation and detection to selectively excite the green or yellow fluorescent protein (Figure 1A).”

7. As mentioned in the section above, I cannot find how long the authors waited after the cranial window surgery until they imaged, but if it is less than four weeks, they need to comment on the effects of inflammation on their synaptic results. This is critical for the interpretation.

We performed acute experiments. As such, we performed the imaging directly after implanting the window and sacrificed the mouse after the session. We performed the cranial window procedure as atraumatic as possible and did not observe tissue damage.

We added: … “mice were anesthetized and a cranial window was implanted above the visual cortex. Imaging commenced about 2.5 hours after onset of the anesthesia.”

8. In lines 287-289, the authors state that bigger spines tend to get smaller and smaller spines tend to get bigger. Given that there is a limit on spine head size, I think that the default hypothesis would be that this reflects regression to the mean. I am not sure why the authors have included this analysis, but they should either show controls that indicate it is not regression to the mean or remove this analysis from the manuscript.

The reviewer addresses here a valuable point. When small spines tend to increase and bigger spines tend to get smaller this reflects indeed regression to the mean and is a common statistical phenomenon. However, this does not affect the interpretation of our data because we compare the changes between Ctr and EE housed mice for the same time intervals and for the whole population of spines. Moreover, we always analyze the whole population and do not group spines into big or small. Analyzing changes exclusively in small or large spines would indeed result in an artifact due to regression to the mean.

In detail, the linear regression such as plotted in Figure 4A–E could be also regarded as a measure for the regression to the mean. The more it diverges from the line of unity, the stronger is the regression toward the mean. As such our data are indeed a measure for the regression toward the mean which is significantly different for PSD95 between EE and Ctr housed mice (Figure 4F).

In the old line 287-289 we refer to our old Figure S5 (new Figure 4—figure supplement 1). In this figure we are plotting the size changes instead of the absolute values and thus it is just another visualization of Figure 4. We would prefer to retain the figure because it links our results to other papers on synaptic size dynamics, e.g. Figure 2 in (Statman et al., 2014) or (Ziv and Brenner, 2018).

We added to the results: “Such a tendency is often called regression to the mean and is an often observed statistical phenomenon. However, it is driven by biological processes, and the strength of those changes may vary under different conditions such as between EE and Ctr. To quantify these changes in synapse and spine head size we use a Kesten process …”

9. As stated in the section above, it is not clear to me the biological relevance of changes in nanoorganization of PSD95. What are the biological consequences or significance of a shift in the nanoorganization for the function of the synapse? Also, could this analysis be quantiative, rather than just descriptive?

As stated in point 2 above, we hypothesize that changes in the PSD95 nanoorganization may cause changes in the alignment of glutamate receptors with the presynapse and thus influence synaptic strength.

The PSD95 nanoorganization is very complex; we often observe clusters but also continuous structure of horseshoe or more complex shapes (See Author response image 2). Previous studies have analyzed numbers of nanodomains (Hruska et al., 2018) which would be possible with relatively simple routines. However, we think that this does not satisfactorily reflect the complexity of the structure. In the future we will develop a shape analysis tool to quantitatively analyze such nanoorganizations. We added all image sections of nanoorganizations to the supplement (Figure 5–source data 1) so that the reader can get an impression about the diversity of shapes.

We included a comment on this issue to the discussion and results.

10. Lines 346-347, what does subtle change or strong change mean for a PSD95 morphology? Can this be quantified as a percentage change of some type? Could the authors also please explain the biological significance or consequences of this change?

We have included more details to the introduction and discussion.

11. In lines 457-458 and 472-473, the authors should cite the original paper that showed that in vivo scaling is input specific, Barnes et al. 2017 (PMID 24395770), not the review that they have cited here.

We have exchanged the reference.

Reviewer #3 (Recommendations for the authors):1) The data quality is amazing, it is very impressive that this resolution is possible in a breathing animal with a beating heart, using relatively slow scanning microscopy. You should mention the complex procedure you developed to ensure stability, normal orientation and biocompatible surface of the cranial window, pointing to the Methods paper for detail. You have earned your bragging rights.

We are happy to read that the reviewer appreciates our superresolution in vivo images. The preparation is indeed the result of years of development.

We added to the results: “To perform motion and aberration free imaging at nanoscale resolution the cranial window needs to be of highest quality. As described in detail in (Steffens et al., 2020) critical steps involve a craniotomy which is as atraumatic as possible, a negligible small gab between brain surface and cover glass and the right choice of the dental cement to avoid bending of the cover glass.”

2) What kind of anesthesia was used during imaging? How much time elapsed between the onset of anesthesia and the first imaging time point (t = 0 min)? Were the imaging experiments performed blind with respect to the housing conditions?

We used MMF, a mixture of fentanyl, midazolam, and medetomidin for imaging and cranial window preparation (cf. Method section).

We added the time between onset of the anesthesia and start of the imaging to the result section: “Imaging commenced about 2.5 hours after onset of the anesthesia.”

The imaging was not performed blind. We ordered the mouse from the animal facility and thus knew which one it was. However, Ctr and EE housed mice were imaged in random order and the dendrites were picked randomly. The analysis was performed blindly. We added these details to the methods section.

3) Lines 235-238: "This means..." This sentence is confusing, delete. The correlation is clearly described in the next sentence. Same for the figure title: "positively and negatively correlated" - I see only weak positive correlations on the population level.

We agree that the sentence “This means…” is partially redundant with the subsequent sentence and therefore deleted it. We also agree that the correlation is positive on the population level and changed the figure title.

In general, Fig. 3 is a bit confusing due to the separate analysis of 3 time points, but no discussion about what happened at t = 0 (onset of anesthesia?). Therefore, the reader is left wondering if the fact that correlation and variance are more or less tight at different time points carries any biological relevance, or if these are supposed to be repeated measures of a Kesten process at work, or perhaps a control for stable imaging conditions? If this is about detecting differences between EE and control, wouldn't it make more sense to pool all time points?

We thank the reviewer for pointing out that it is unclear why we performed the time lapse for the three time intervals and what we expected. T = 0 marks the first imaging measurement about 2.5 hours after onset of anesthesia and cranial window preparation (we clarified that cf. remark 2).

Meyer et al (Meyer et al., 2014) have shown that the brightness of fluorescently labeled PSD95 increased slowly over 180 min after glutamate uncaging. Bosch et al. (Bosch et al., 2014) found that the postsynaptic scaffolding proteins Homer1b and Shank1b persistently increased for up to 150 min after LTP. Thus we hypothesized that we would find larger changes in PSD95 area after 2 hours (Fig. 3F) than after 30 min (Fig. 3D). The most straightforward way to analyze these changes is to compute the variance of the changes. Fig. 3 D–J shows that we did not find a statistically significant change in variance between the different time intervals for each group. This changed when we plotted these changes specifically over the original size (Figure 4–figure supplement 1) or the size at t+Δt over the size at t as shown in Fig. 4. In this way, we found strong differences in the slope of the PSD95 area changes (Fig. 4F). The slope is one of the two parameters which are used as variables in the Kesten process (nicely explained in (Ziv and Brenner, 2018)). Previous experiments and simulations of the Kesten process have shown that a decrease in this slope, termed multiplicative downscaling, results in a narrower synaptic size distribution which is exactly what we observe for PSD95. We have tried to emphasize this aspect more in the manuscript. Thus, we found indeed differences in the temporal changes between different time points (Fig. 4), but they are not visible in the simple analysis of variance (Fig. 3).

However, we took up the last suggestion of the reviewer and pooled the variances for the different time intervals per group. The variance of the Ctr group is larger for both PC1 and PC2, however, this difference is highly significant only for PC2.

Therefore, we added: “This means that the negatively correlated changes contribute much less than the positively correlated changes in the EE housed mice indicating a stronger, positive coupling between changes in spine head size and PSD95 area.”

We add panel G and J to Fig. 3.

4) Fig 4F, control animals: I have a hard time understanding how there can be shrinkage at 30 min sampling intervals, but not at longer intervals. Does the dt30min group only contain (30 min - 0 min), or also (60 min - 30 min)? Does this mean initially shrunken PSDs grow back again? Has this something to do with the onset of anesthesia? Please explain/interpret this result.

A small misunderstanding may have occurred here. A small value for the slope does not necessarily mean shrinkage. The linear regression lines for all measures (Fig. 4A, B, D, E) have a positive y-intersect and a slope <1 which indicates that small spines preferentially grow while large spines preferentially shrink. The average value is roughly stable (Fig. 3C). Thus, we observe a size dependent, multiplicative component which is indicated by a slop ≠1 and an additive component indicated by a positive y-intersect. This fits to a Kesten process which is a statistical framework for a stochastic process combining a multiplicative downscaling, and an additive growth (Statman et al., 2014); we have performed measurements at baseline, i.e. steady state, at which the average size is constant.

However, we cannot fully explain this increase in slope for the Ctr mice. Previous experiments have shown a steady decrease of the linear regression slope over time as shown in (Hazan and Ziv, 2020)(Statman et al., 2014). However, these studies were performed in cultured neurons over a very long temporal range of up to 50 hours, by using overexpression of PSD95-GFP, and the fluorescence brightness was taken as a measure for size. The measurements were smoothed over different time points, precluding an analysis between consecutive time points / short time intervals. It is also conceivable that such highly regulated changes follow a different time line in vivo. In the future we will use our recently established chronic STED imaging (Steffens et al., 2021) to follow the PSD95 size dynamic also on longer time scales to analyze the long term dynamic such as in (Statman et al., 2014).

We have refined the discussion section “Stronger multiplicative downscaling in EE housed mice for PSD95”

Imaging was performed about 2.5 hours after the onset of the anesthesia and thus should be in a stable regime regarding the anesthesia. We added this information to the result section.

We indeed pooled the data, i.e. for the 30 min interval, we included 0–30 min and 30–60 min changes. The rational to do that is that we measured at baseline, i.e. we did not stimulate and measured long after onset of the anesthesia. However, to show that pooling did not affect the outcome of the study, we also analyzed the first interval only (0–30 min, 0–60 min and 0–120 min). As shown in Author response image 5, the first interval only provided similar results as the pooled data, just with larger error bars:

**Author response image 5. sa2fig5:** Slope of the linear regression as shown in the manuscript Fig.4C, F for pooled data and first time interval only. The pooled data includes 0–30 min and 30–60 min data for Δt = 30 min. Δt = 60 min includes 0–60 min and 60–120 min of the hourly measurement interval and 0–60 min of the half hour measurement series. Pooled data for Δt = 120 min includes 0–120 min of the 2 hour time interval measurement and 0–120 min of the hourly measurement series. For comparison, we computed the linear regression for the first interval only, which are 0–30 min, 0–60 min and 0–120 min (right). There were no major differences between the pooled data (left) and the data for the first interval (right), except for a much larger error bar for the 120 min interval..

We added a comment on the pooling to the legend of Fig. 3: “Time intervals are pooled; e.g. Δt = 30 min includes 0–30 min and 30–60 min.”

5) Fig. 4G: This analysis is great, but its significance might be difficult to understand for some readers. It might be worth pointing out that if there were a temporal sequence, e.g. first spine size expansion, then PSD enlargement, this would result in little correlation when comparing just two time points. Thus, the cross-correlation analysis. One could even do a little simulation to illustrate how the cross correlation would look like if changes were linked with a delay (this is optional).

We agree and thank the reviewer for pointing this out. We have separated this paragraph for clarity and added a motivation:

“No temporal shift between spine head and PSD95 area changes at baseline

We found a strong correlation coefficient of ~0.8 between spine head and PSD95 area (Figure 2B). However, if the PSD expands with a temporal delay of ~1 hour to the spine head as suggested by the work of Bosch et al. and Meyer et al. (Bosch et al., 2014; Meyer et al., 2014), the correlation should be even higher when comparing spine heads and PSD95 area at different time points. Therefore, we computed the cross-correlation between these measures for all time intervals.”

6) Did some spines disappear completely during the period of observation?

We thank the reviewer for pointing out this interesting question. We saw indeed some protrusions appearing and disappearing during the observation period. All of these new or lost protrusions, however, did not bear PSD95 puncta and did not show a thickening at the end, a spine head. Thus these highly mobile spines were most likely filopodia. We observed such a new or lost filopodium in every second to third image. These are not enough data points for a size analysis or detailed description of filopodia turnover.

We added this observation to the results: We recorded STED images at different fields of view; each field of view was recorded at three time points at a time interval Δt of either 30 min (Fig. 3A, B), 60 min or 120 min. “Over these time periods the spines were mostly stable. Occasionally, a spine was lost or a new one appeared; none of these spines carried PSD95 and they were therefore most likely highly dynamic filopodia (Berry and Nedivi, 2017).”

[Editors' note: further revisions were suggested prior to acceptance, as described below.]

The manuscript has been significantly improved but there are some remaining issues, mostly concerning aspects of the biological context of the study, that need to be addressed, as outlined below:

We would like to thank the reviewer for the very helpful comments that improved the clarity of the manuscript. We have addressed all points raised below and hope the revised version of our manuscript is now suitable for publication.

Line 32-33: "… synaptic strength is set more precisely."The term "precise" implies that there is a specific value around which the synaptic strength is set, which is not clear in this case. It is suggested that the authors use a more neutral expression such as ".. synaptic strength is set more uniformly" to describe the key observation.

We agree and changed the wording accordingly. (New line 28)

Line 62: "… which was confirmed recently (Holler et al., 2021)."It would be informative to indicate that the study involved EM (CLEM) analysis, and should be rephrased, for example, "… which was confirmed recently at ultrastructural resolution…".

Thank you, we added this phrase. (New line 52)

Line 67: "… directly after LTP."-> … directly after LTP induction.

Added. (Line 56)

Line 70: "… with a delay of ~1 hour after LTP"As with above, potentiation itself can last for hours, and thus the timeframe being referred to needs to be clear.-> e.g. … with a delay of ~1 hour after inducing LTP

This is indeed a valuable point. We added “inducing” (Line 58)

Line 76: "… of the spine and postsynapse at increased activity and baseline in vivo."The increased activity is an assumption, and this should be reflected in the statement. For example, one could rephrase along the lines of "… of the spine and postsynapse in vivo using mice reared in enriched environment representing increased activity conditions and normal housing representing baseline conditions."

We agree that this is more specific and changed the sentence as suggested. (Line 63-65)

Line 123Is it meant that in one possibility, EE could directly affect "pre-existing" synapses rather than "all" synapses?

In our experimental setting, we cannot distinguish between spines that were affected by EE and those that were not. However, with this sentence (New line 91-92) we wanted to emphasize that previous studies focused on the observation of spine stability, thus spine formation and elimination, but nothing is known about morphological changes of the spines and synapses. Such morphological changes could affect all spines – those newly formed due to the EE housing and preexisting. Of course, in the future it would be interesting to investigate whether the morphology of spines and synapses which are formed due to learning are different than the preexisting ones.

We rephrased as follows: “However, it is unknown whether the effects of EE leave their mark only at the level of spine formation and elimination such as observed by Yang et al. (Yang et al., 2009), or whether EE affects also the dynamics and nanostructure of all individual spines and PSDs.”

Line 190In addition to the reference to Steffens et al., 2020, it would be helpful to the reader if the authors briefly mention what particular procedural features help ensure the craniotomy be atraumatic as possible.

With the phrase “as atraumatic as possible” (Line 155) we do not refer to a particular procedure. We want to emphasize that care needs to be taken at all surgical steps, the drilling, removal of the bony plate and removal of the dura to not damage the cortical surface. This may seem obvious, but in our experience, different experimenters perform a craniotomy very differently. This is much more critical for superresolution microscopy than for two-photon imaging but difficult to put into stringent operating instructions.

We added more details: “As described in detail in Steffens et al. (Steffens et al., 2020), critical steps involve a craniotomy that is as atraumatic as possible and does not damage the cortical surface when drilling or removing the bone plate and dura mater. Moreover, the gap between the brain surface and the cover glass needs to be negligible small and the right choice of the dental cement is important to avoid bending of the cover glass.”

Line 580: "To study activity-dependent changes, we compared adult mice housed in standard cages with age matched EE housed mice and analyzed synaptic morphology and plasticity …."The motivation for the study is somewhat misguided and unclear, since the present work assessed the influence of activity by subjecting mice to activity-enhanced conditions by means of EE, and did not directly examine activity-dependent changes per se. Moreover, as acknowledged by the authors, there may be an issue also of anesthesia. The starting sentence as well as the rest of the discussion in the paragraph need to reflect these points.

We thank the reviewer for this comment. Indeed, we did not measure activity-dependent changes but changes due to the different rearing conditions – with and without enhanced activity.

We rephrased the first sentence accordingly (line 489): “To study whether different activity conditions during rearing influence synapse and spine size and plasticity, we compared adult mice housed in standard cages with age matched EE housed mice and analyzed synaptic morphology and plasticity in the visual cortex after the critical period.”

A large part of the paragraph deals mainly with spine and PSD95 assembly size differences between EE and Ctr housed mice. These sizes should not be influenced by the anesthesia and thus reflect differences between EE and Ctr housing as described.

At the end of the paragraph, where we discuss “the correlation between spine head and PSD95 changes”, we added a note about the influence of the anesthesia: “As discussed above, however, the dynamic might be influenced by the anesthesia and different in the awake state. (Line 511).”

It should be noted that both groups, EE and Ctr, were treated in the same manner and imaged under anesthesia. Therefore, we assume that the difference in plasticity is due to the different rearing /housing conditions with and without enhanced activity. However, the amplitude of those changes might indeed be different in the awake state.

Paragraph starting from Line 606As with the comment above, the caveats of the present experimental design, including the effects of anesthesia, should be acknowledged.

In this paragraph, starting in new line 513, we discuss the dynamical changes and we agree with the above comment of the reviewer that it should be mentioned that these changes do not reflect activity dependent changes at the moment of the measurement but do reflect the plasticity after rearing under enhanced activity.

Thus, we added (Line 525): “However, it should be noted that our in vivo measurements were performed under anesthesia and directly after implanting a cranial window; therefore differences are due to the different rearing conditions and not to changes in activity at the moment of the measurement. The in vitro silencing, in contrast, continued over the measurements.”